# Spatially resolved integrative analysis of transcriptomic and metabolomic changes in tissue injury studies

Eleanor C. Williams [1,2], Lovisa Franzén [3,4], Martina Olsson Lindvall [3], Gregory Hamm[2], Steven Oag[5], Muntasir Mamun Majumder [3], James Denholm[2,6], Azam Hamidinekoo[7], Javier Escudero Morlanes [4], Marco Vicari[4,8], Joakim Lundeberg [4], Laura Setyo[7], Trevor M. Godfrey [9], Livia S. Eberlin [9], Aleksandr Zakirov[10], Jorrit J. Hornberg[3], Marianna Stamou[3], Patrik L. Ståhl [4], Anna Ollerstam[3] ✉, Jennifer Y. Tan[11] ✉ & Irina Mohorianu [1] ✉

Recent developments in spatially resolved -omics have enabled the joint study of gene expression, metabolite levels and tissue morphology, offering greater insights into biological pathways. Integrating these modalities from matched tissue sections to probe spatially-coordinated processes, however, remains challenging. Here we introduce MAGPIE, a framework for co-registering spatially resolved transcriptomics, metabolomics, and tissue morphology from the same or consecutive sections. We show MAGPIE's generalisability and scalability on spatial multi-omics data from multiple tissues, combining Visium with MALDI and DESI mass spectrometry imaging. MAGPIE was also applied to new multi-modal datasets generated with a specialised sampling strategy to characterise the metabolic and transcriptomic landscape in an in vivo model of drug-induced pulmonary fibrosis and to link small-molecule co-detection with endogenous lung responses. MAGPIE demonstrates the refined resolution and enhanced interpretability that spatial multi-modal analyses provide for studying tissue injury especially in pharmacological contexts, and delivers a modular, accessible workflow for data integration.

Recent advances in spatially resolved -omics technologies have enabled untargeted studies of spatial distributions of various molecular species[1,2]. Spatially resolved transcriptomics (SRT) enables genome-wide spatial gene expression profiling[3–8], allowing the untargeted exploration of disease-driving mechanisms at an unprecedented scale and resolution[9–13]. To further evaluate disease- and drug-induced alterations in situ, it is highly valuable to capture multi-modal data[14]. The combination of SRT and mass spectrometry imaging (MSI)[15,16] in the same tissue specimen has expanded the limits of spatial multi-modal analyses[17–20], facilitating the study of smaller, endogenous and/

[1]Cambridge Stem Cell Institute, University of Cambridge, Cambridge, UK. [2]Integrated Bioanalysis, Clinical Pharmacology & Safety Sciences, R&D, AstraZeneca, Cambridge, UK. [3]Safety Sciences, Clinical Pharmacology & Safety Sciences, R&D, AstraZeneca, Gothenburg, Sweden. [4]Department of Gene Technology, KTH Royal Institute of Technology, Science for Life Laboratory, Stockholm, Sweden. [5]Animal Science and Technology, Clinical Pharmacology and Safety Sciences, R&D, AstraZeneca, Gothenburg, Sweden. [6]Department of Radiology, University of Cambridge School of Clinical Medicine, Cambridge, UK. [7]Pathology, Clinical Pharmacology and Safety Sciences, R&D, AstraZeneca, Cambridge, UK. [8]Unit of Integrative Metabolomics, Institute of Environmental Medicine, Karolinska Institutet, Stockholm, Sweden. [9]Department of Surgery, Baylor College of Medicine, Houston, TX, USA. [10]Department of Clinical Neurosciences, University of Cambridge, Cambridge, UK. [11]Predictive AI & Data, Clinical Pharmacology & Safety Sciences, R&D, AstraZeneca, Cambridge, UK. ✉e-mail: anna.ollerstam1@astrazeneca.com; jennifer.tan0102@gmail.com; iim22@cam.ac.uk

or exogenous molecules and metabolites, and opening new avenues for functional characterisation of local compound-induced tissue responses. While parallel unimodal analyses offer valuable insights, fully integrated spatial multi-omics workflows allow the direct investigation of cross-modality covariation within the same microenvironment and histological context. To unlock such analysis, it is essential to align all modalities into a shared coordinate system. However, generating multi-modal data for successful co-registration and co-analysis relies on both experimental and computational considerations.

On the experimental front, there are two established approaches: (1) developing protocols that allow multiple measurements on the same tissue section[18] or (2) collecting each data modality on separate, consecutive tissue sections[17,19,20]. In both cases, sample preparation requirements ensure high analyte quality and compatibility with each modality. A typical approach for SRT would include optimal cutting temperature (OCT) tissue embedding; however, this obstructs MSI data collection[21,22], prompting more appropriate methods to be considered for joint SRT and MSI analysis. Furthermore, refined methods may allow the addition of extra modalities for the assessment of tissue morphology, as captured by default in the Visium protocol and optionally obtainable after MSI data acquisition by histological staining and imaging of the tissue section.

Computationally, spatial analyte abundance levels and images obtained for each modality need to be co-registered to enable direct comparisons. Many current methods use 'landmarks' (corresponding locations across different images), identified either manually or automatically[23]. Once landmarks are identified from histological or spatial expression patterns, the geometric transformations required to align multiple images are estimated and applied. Given the inherent differences between consecutive sections and potential distortions introduced during sample preparation, straightforward linear mappings may fail to achieve accurate alignments and lead to compromised interpretations; more complex, non-linear, and elastic transformations may thus be required to effectively link corresponding spatial features across images. Several methods have recently been developed for this co-registration process, including SpatialData[24] and Giotto Suite[25], which offer purpose-built frameworks with data structures designed for spatial -omics, coupled with functions for alignment and aggregation of observations across modalities, including Visium and others. Nonetheless, these toolkits lack full interoperability with the existing landscape of spatial tools, such as R-based and multi-omics methods[26–29] and further lack specific support for MSI data. SOmicsFusion[30] supports MSI data, although it focuses on imaging data with no support for Visium datasets and is currently solely provided as standalone scripts, which may limit the ease of use for some users. STalign[31] and Eggplant[32] enable the alignment of multiple SRT datasets, however, they do not specifically support MSI data and have not been tested in the multi-omics setting. Across these existing methods, we noted a distinct lack of tools that perform standalone, accessible, and interoperable co-registration of spatial modalities, with particular support for combining sequencing-based and imaging-based modalities such as Visium and MSI. Such interoperability is essential to access existing and upcoming downstream analysis options that require matching spatial coordinates and observations[28,29,33–35]. Additionally, integrating MSI with Visium is complicated by their differing spatial resolutions, which can obscure true molecular co-localisations, motivating specialised approaches to avoid resolution-driven artefacts.

To overcome these challenges, we present a streamlined open-source computational workflow, MAGPIE (**M**ulti-modal **A**lignment of **G**enes and **P**eaks for **I**ntegrated **E**xploration), for integrating spatial transcriptomics (Visium) and metabolomics (MSI) datasets, alongside histological images, generated on same or consecutive sections. MAGPIE outputs standardised and readily usable files to be processed by downstream Python- or R-based spatial toolkits, making it versatile

and scalable towards a wide variety of further computational spatial multi-omics analytical methods. The generalisability and scalability of MAGPIE was benchmarked on several datasets, covering different tissue types and MSI technologies, including publicly available brain and breast cancer datasets. To illustrate the full versatility of the framework, we generated and processed spatial multi-modal datasets in two tissue injury studies using lung tissue. As the lung is an inherently heterogeneous tissue type, requiring specific care to preserve its integrity, we developed a new sample preparation strategy to augment compatibility with both SRT and MSI analyses.

In summary, MAGPIE showcases the capacity of efficient spatial multi-modal data integration to reveal greater insights into disease-driving mechanisms and to enhance the study of local drug-induced alterations within a tissue.

## Results

### Integration of spatially resolved transcriptomics and metabolomics with MAGPIE

To enable and benefit from full spatial multi-omics integration, each modality should be generated from the same tissue specimen. This may require appropriate adaptations of the experimental protocols for each modality and tissue type. Specifically, to enhance the quality and subsequent interpretability of spatial multi-omics datasets from Visium and MSI in lung tissue, a highly heterogeneous and fragile tissue type, we developed a new experimental sampling approach (**Methods**). Inflation of fresh rodent lung tissue using an agarose solution followed by direct freezing allowed preservation of tissue morphology while avoiding the need for OCT embedding, hence facilitating MSI analyses in tissue sections adjacent to those analysed with Visium to provide matched spatial multi-modal data (Fig. 1a, and Supplementary note 1).

For the integrated computational analysis of spatial multi-omics assays, with a focus on transcriptomics and metabolomics, we developed the MAGPIE pipeline (Fig. 1b), implemented as a Snakemake workflow. As input, MAGPIE takes standard Space Ranger (10x Genomics) outputs from Visium acquisition and MSI data processed through standard software (e.g., SCiLS Lab or Cardinal[36]) into a tabular peak-by-pixel format. To co-register the MSI data with the image acquired from the Visium workflow, an MSI-representative image is generated in MAGPIE through dimensionality reduction of the MSI intensity data, allowing for the identification of tissue boundaries and morphological features. For this step, the user may guide the process to include a priori knowledge of spatially informative peaks or to colour each pixel based on 1-3 principal components (visualised as RGB channels). To support the alignment, the pipeline also accepts an intermediate tissue image for the MSI modality, such as a microscopy image of the tissue section, preferably stained similarly to the matching Visium section, e.g., with haematoxylin and eosin (H&E) or with immunofluorescence (IF). In this case, co-registration is first performed between the MSI data-generated image and the MSI tissue image, and thereafter between the MSI tissue image and the Visium tissue image. To manually select spatial landmarks for co-registration, MAGPIE includes an interactive (Python Shiny) application developed for this purpose. Alternatively, users can opt to incorporate externally identified landmarks, such as those generated using the deep learning-based unsupervised landmark detection method, ELD[23], providing flexibility in the landmark selection process.

Once landmarks are identified, transformations of the queried MSI image to the reference Visium image can be performed using linear (e.g., affine) or non-linear (e.g., thin plate splines (TPS)) transforms. The transformation alignment projects the two modalities into a common coordinate framework (CCF), such that an (x,y) coordinate in one modality (MSI) can be directly mapped onto the other modality (Visium). Thereafter, the pipeline prepares the aligned second modality (MSI) data for outputting, by converting all data into the

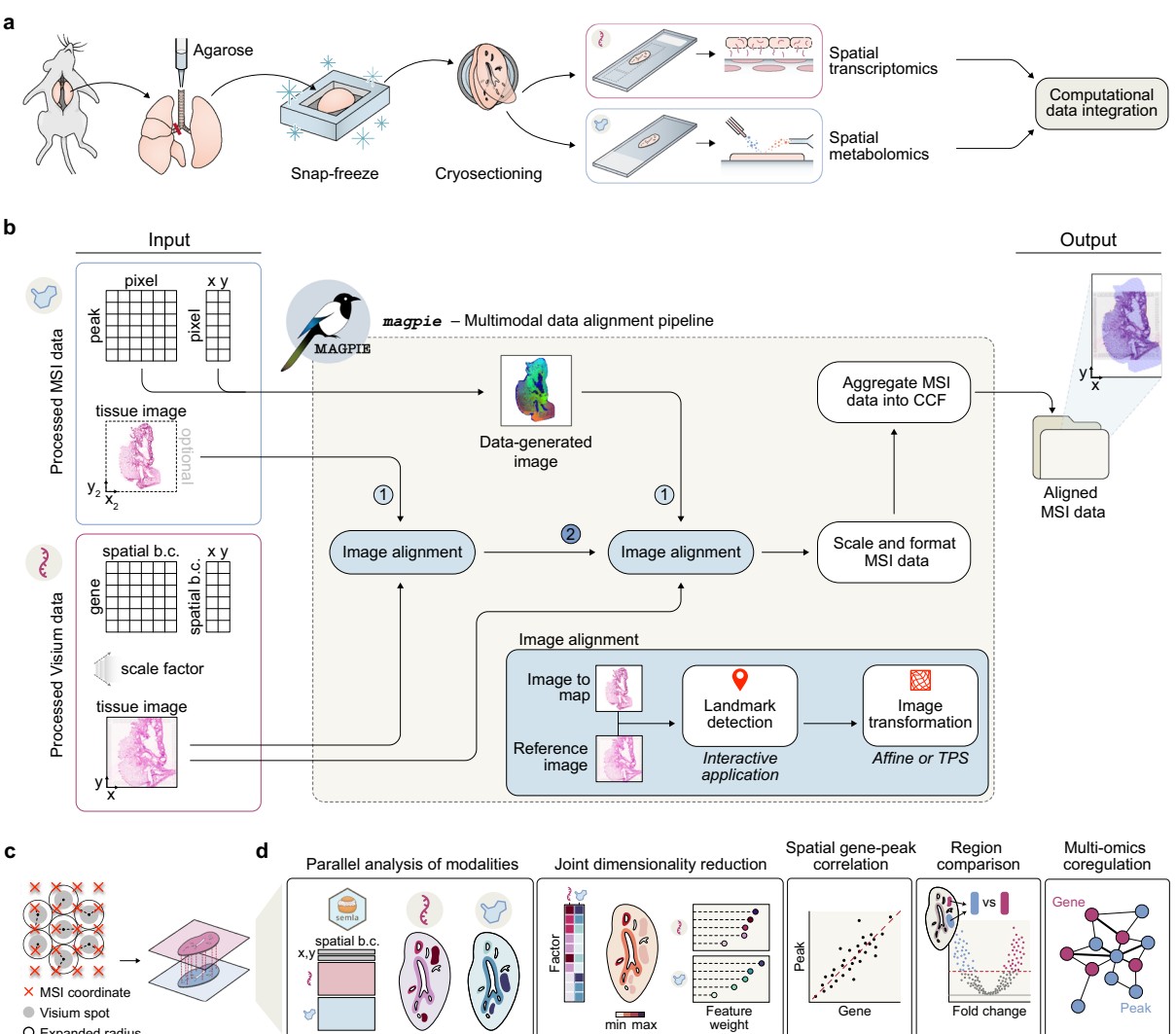

**Fig. 1 | Spatial multi-modal processing workflow and integration using the MAGPIE framework. a** Experimental workflow for spatial multi-omics profiling of rodent lung, consisting of agarose inflation of the lung tissue to facilitate the preservation of tissue integrity, followed by snap-freezing and cryo-sectioning to create consecutive sections, used for spatial transcriptomics and spatial metabolomics, respectively. **b** MAGPIE computational framework for co-registering same or consecutive section spatial transcriptomics (Visium) and metabolomics (mass spectrometry imaging, MSI) data. The pipeline's inputs and outputs are in standardised Space Ranger-style and tabular formats to ensure compatibility with other tools. Preprocessing the MSI data produces a data-generated image used for landmark selection and subsequent image co-registration. The output from the pipeline is the MSI data with updated coordinates aligned to the Visium data. **c** To create a 1:1 mapping between Visium spots and MSI pixels, MAGPIE (by default) expands Visium spot radii, such that there are no gaps between spots, and then aggregates MSI pixels that fall within these spot boundaries. This results in matching observations between modalities and a processed object which can then be read by analysis toolkits, including semla[26]. **d** Overview of downstream analysis options once the modalities are aligned into a matching coordinate system. Examples include joint clustering, dimensionality reduction, linking transcriptomic and metabolic changes to histologic information, and gene-peak correlation or multi-omics covariation network analysis. Spatial b.c. spatial barcode, CCF common coordinate framework. Source data are provided as a Source Data file.

standardised Visium Space Ranger-style output (i.e., MSI 'count' data in .h5 file format and coordinates, scale factors, and the transformed tissue image into a 'spatial' folder), which is readily accessible for downstream analysis using several spatial analysis ecosystems, such as semla[26], Squidpy[37], or others[24,27,38,39]. To perform a fully integrative analysis, pixels and spots are further combined into common 'observations', to ensure a matching at 1:1 resolution across the two modalities. To achieve this, MAGPIE by default maximally expands the radius of each Visium spot, while avoiding intersections with the neighbouring expanded spot radii. Alternatively, it uses the actual Visium spot size (55μm) or an estimated MSI pixel size according to user choice. It thereafter aggregates (by sum, mean, or weighted average) the values from all MSI pixels mapped within the selected spot radius (Fig. 1c, and Supplementary note 2). The MSI pixel

observations are thereby assigned to the same coordinate identifiers as used for the Visium spots, resulting in matching spatial resolutions between the aggregated MSI and Visium data. Despite potentially lowering the spatial resolution of the original MSI data, we recommend this approach rather than generating computationally inferred super-resolved Visium data to avoid noise issues in MSI data. However, as differences in sampling scale can influence the apparent strength of co-localisation, users should carefully consider the resolution disparity in their own datasets when selecting an aggregation approach and interpreting their results. If the MSI dataset is of significantly higher resolution than its Visium counterpart, where entire MSI pixels may fall in the gaps between Visium spots, we recommend using the true Visium spot size to aggregate data instead of the expanded Visium radius. If the MSI resolution is lower than Visium, it may be useful to focus on

the MSI pixel size for aggregation where one MSI pixel may contribute to multiple Visium spots. In addition, this aggregation approach has been added as a new function within the semla R package[26] (v. ≥ 1.3.0) to allow the creation of a spatial multi-modal object and further analysis within the Seurat or semla R ecosystems.

MAGPIE is a straightforward, optimised framework which is fast and lightweight to run, as demonstrated by run time and computational requirement benchmarking for various MSI dataset resolutions (Supplementary Fig. 1a). Once the data is fully integrated, a breadth of downstream spatial multi-modal processing, analysis, and visualisation options become possible (Fig. 1d), facilitating discoveries on joint metabolomic and transcriptomic dynamics in healthy, diseased, or other perturbation settings.

## Data-driven guidance for enhanced MAGPIE co-registration

To ensure the successful co-registration of Visium and MSI datasets using MAGPIE, both technical and experimental factors need to be considered. In particular, we explored the effect of sample quality and parameter selection on co-registration performance, both of high importance when working with consecutive tissue sections where the section similarity is greatly affected by the sectioning process and tissue quality.

Several decision juncture points along the MAGPIE pipeline can impact the success of the resulting co-registration (Fig. 2a). Specifically, these user decision points are (i) whether a microscopy image for the second modality is generated or if co-registration is performed directly from the data-generated image, where successful landmark identification can be more difficult, (ii) the colouring used for the data-generated image, which may impact the ease and accuracy of landmark identification, (iii) how many landmarks are used and (iv) whether an affine or TPS transformation is used to map between modalities, of particular importance when the tissue sections used for each modality differ strongly. To assess these factors and guide the study design and parameter settings when running MAGPIE, we generated DESI MSI data and performed H&E staining of consecutive sections of mouse lung tissue (n = 11) previously analysed with Visium[13].

Following the standard pre-processing of each modality for our generated paired Visium and MSI data, the data was processed with MAGPIE to benchmark the impact of altered input parameters. To assess the extent of co-registration success, we computed an alignment accuracy score based on the overlap between tissue and background labels in the Visium and MSI observations, specifically by comparing the number of spots labelled consistently in both modalities after transformation to the total number of spots (**Methods**). For the successfully co-registered samples (8 out of 11), up to 20 distinct landmarks per sample were manually selected using the interactive application within MAGPIE, allowing the assessment of alignment accuracy as a function of the number of landmarks (Fig. 2b, and Supplementary Fig. 1b). Regardless of transformation type or the inclusion of an intermediate H&E image for the MSI modality, the accuracy score started to plateau at around 10 landmarks, with small additional improvements for some samples when including more landmarks (Supplementary Fig. 1b). To better understand the differences in co-registration between samples, we compared the maximum achieved alignment accuracy, across up to 20 landmarks, between samples with or without an intermediate H&E image and across image transformation types (affine vs TPS) (Fig. 2c). Compared to direct co-registration from MSI data to Visium, co-registration using an intermediate MSI H&E image attained higher alignment accuracy in terms of tissue overlap, with a larger difference observed when a TPS transformation was applied (Fig. 2d).

The co-registration accuracy differences across samples (Fig. 2c) underline extensive discrepancies between the sample section pairs,

which may be explainable by the morphological distortions present within the H&E-stained tissue sections (Fig. 2e). The samples with the highest maximum accuracy were characterised by well-preserved tissue integrity, with only minor tears and folds (Supplementary Fig. 1c). In cases where a section was too severely damaged, co-registration using manual landmarks was not possible (e.g., sample SX), while relatively high alignment accuracies could be achieved with MAGPIE where there were moderate tears and shape differences in the section (e.g., sample S8). Some of the more dissimilar samples demonstrated greater accuracy differences when intermediate MSI H&E images were utilised, highlighting their added value.

Based on these experiments and benchmarking results, we strongly recommended obtaining an intermediate MSI image (e.g., H&E staining after MSI) for more robust landmark identification and enhanced co-registration. In addition, for cases where landmark identification is challenging, opting for a linear transformation, such as affine, can help to avoid exaggerated distortion of the image transformation potentially caused by improperly placed landmarks. Conversely, for more dissimilar serial sections, a non-linear transformation may be necessary to achieve high-quality co-registration. Finally, an added benefit of selecting a high number of landmarks to aid the co-registration was observed, and we, therefore, recommend that users aim for higher numbers of manually selected landmarks.

## MAGPIE provides flexibility across tissue types and technologies

Next, we showcased the versatility of MAGPIE on samples from different species, tissue types, and MSI technologies. Co-registration was performed on previously published same-section Visium and MALDI MSI data from human (n = 3) and mouse (n = 3) brain tissue[18] (Spatial Multimodal Analysis or SMA dataset) (Fig. 2f, and Supplementary Fig. 1d). We performed co-registration on samples processed in both positive and negative ionisation modes and using different matrices (FMP-10, 9-AA, DHB) to showcase the versatility of the MAGPIE framework. Testing up to 15 landmarks for each sample, all the included SMA samples achieved near-perfect co-registration accuracy (≥ 0.9), illustrating the successful alignment of datasets and the benefit of having same-section multi-modal data (Fig. 2f, and Supplementary Fig. 1e). Nonetheless, challenges in landmark identification and alignment accuracy assessment were encountered for the human brain SMA samples, which likely stemmed from their complex underlying structure and morphology and the lack of visible tissue boundaries, since the data was acquired from small regions of interest within a larger tissue block. Conversely, the mouse brain SMA samples had distinct morphological features and section edges that eased the manual landmark selection process. Hence, as a consideration, the placement and integrity of the tissue section can aid landmark selection if more distinguishable features are present, where holes and tears may even be beneficial. In addition, the choice of alignment metric can impact the presented co-registration accuracy scores, which in our case was strongly influenced by the tissue-to-background selection and differences between MSI and Visium capture area sizes. Focusing on image-level differences, rather than spot- or pixel-level, might present a complementary alignment accuracy score metric, albeit reliant on having an intermediate H&E image from the MSI modality, which was not available in our case.

Our results show that successful co-registration with MAGPIE is achievable for different tissue types and data acquisition technologies. Further, we explored MAGPIE's ability to resolve highly localised brain structures (Supplementary note 3), demonstrating MAGPIE's generalisability to neuroscience contexts We highlight the importance of carefully selecting samples for multi-omics integration, particularly regarding the clarity of histological features, which are crucial for successful co-registration and especially important when working with larger or more complex tissue sections.

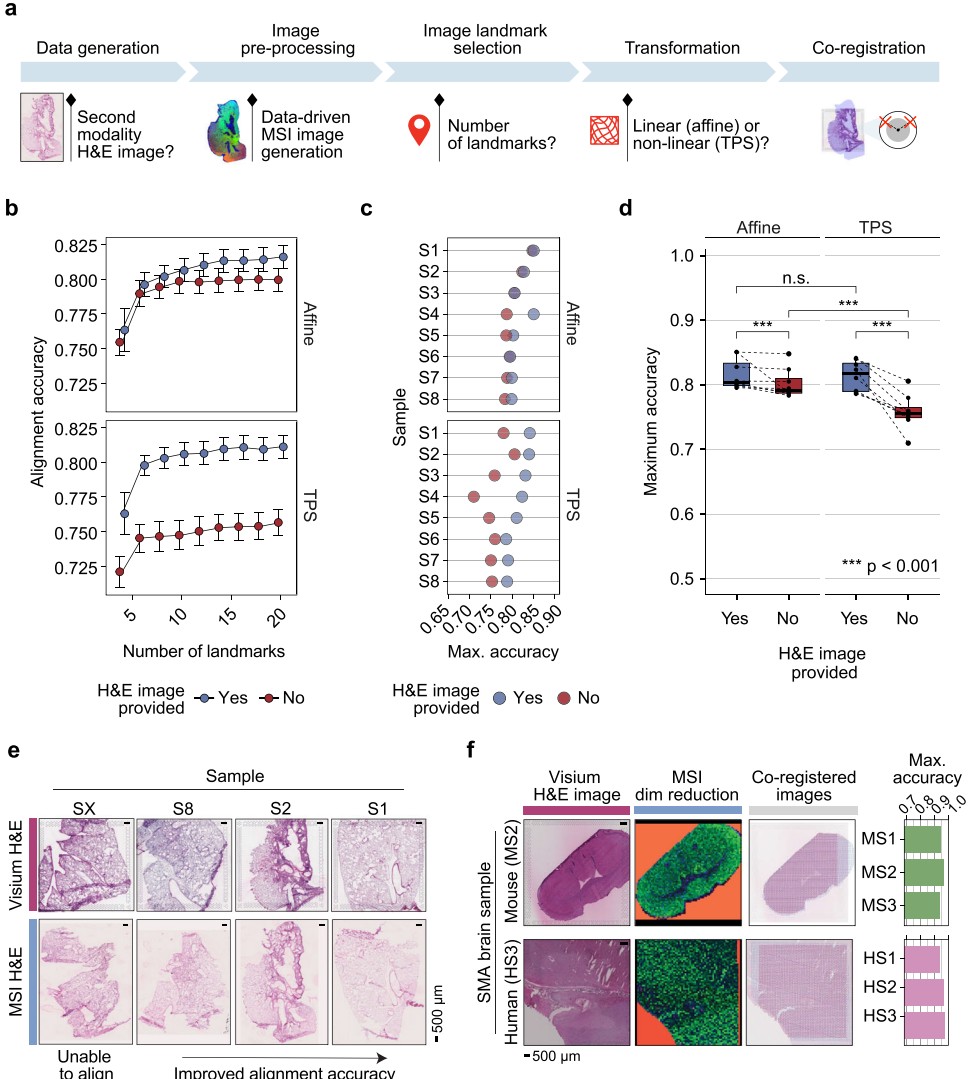

**Fig. 2 | Demonstration of MAGPIE flexibility and robustness. a** Summary of the MAGPIE pipeline key decision points for user consideration. **b** Variation in the alignment accuracy across a range of landmarks, employing a linear (affine) or non-linear (TPS) transform, with or without using an intermediate MSI H&E image to assist with co-registration, for 8 mouse lung samples. For each setting, the mean alignment accuracy per sample was selected across 5 iterations of randomly selected landmarks followed by averaging the scores across all samples, shown as points in the graph. Error bars correspond to the standard error centred on the mean alignment accuracy across all samples. Points are coloured by the inclusion of an intermediate H&E image for the MSI modality. N samples = 8. **c** Maximum accuracy for each sample using either affine or TPS transform, with or without the aid of an intermediate MSI H&E image (reflected in colour). **d** Comparison of highest accuracy performance per sample using affine or TPS transforms, with or without an intermediate H&E microscopy image. Each point represents the maximum achieved accuracy for each tested sample (n = 8) with the boxplot showing the median, upper and lower quartiles with whiskers extending to 1.5× interquartile

range. The maximum alignment accuracies per group were compared using paired two-sided Wilcoxon rank sum tests (Affine with H&E vs Affine without H&E: $p = 1.12 \times 10^{-4}$, TPS with H&E vs TPS without H&E: $p = 1.77 \times 10^{-13}$, Affine with H&E vs TPS with H&E: $p = 0.198$ (*n.s.* non-significant), Affine without H&E vs TPS without H&E: $p = 1.69 \times 10^{-13}$). Boxes are coloured by the inclusion of an intermediate H&E image for the MSI modality. **e** Examples of H&E-stained tissue sections of mouse lung section pairs, ordered by maximum accuracy measurement between Visium and MSI sections. Full results across 9 tested samples with Visium and MSI H&E images are shown in Supplementary Fig. 1c. Scale bars reflect 500 μm. **f** Generalisability of MAGPIE to other species and tissue types by application to same-section Visium and MALDI-MSI data (SMA) from mouse and human brain tissue. The maximum co-registration accuracies achieved per sample (mouse, n = 3; human, n = 3) are shown as bar charts, alongside example Visium H&E images, MSI dimensionality reduction colouring (using first 3 principle components as RGB channels) and overlaid Visium and MSI coordinates. Scale bars reflect 500 μm. Source data are provided as a Source Data file.

## Alignment of spatial multi-omics in pulmonary fibrosis reveals disease-associated signatures

To explore the insights into disease-driving mechanisms provided by MAGPIE-integrated spatial multi-modal data, we further analysed the dataset comprising Visium and DESI MSI on mouse lung samples (introduced in Fig. 2). These samples were collected from a mouse model of bleomycin (BLM)-induced pulmonary fibrosis, a widely used preclinical animal model for studying lung fibrosis and for the identification and assessment of new therapeutic drugs for human

idiopathic pulmonary fibrosis (IPF)[40,41]. The administration of the cytotoxic agent BLM induces acute inflammation and subsequent fibrotic scarring of the lungs[42,43]. Recently, an in-depth Visium-based characterisation of tissue samples from IPF patients and the BLM mouse model was published, describing and comparing molecular signatures associated with the fibrotic niches[13]. To further build on this study and previous reports of metabolic alterations seen in IPF and BLM-induced pulmonary fibrosis[44–46], our spatial multi-modal Visium and DESI MSI data from mouse lungs collected 21 days following BLM

administration (n = 6) were integrated and analysed within their histopathological context along with cell type mapping inference[13,47] (Fig. 3a).

The co-registration of the modalities into a shared coordinate system enabled us to summarise patterns across modalities using a multi-omics factor analysis (MOFA)[29,48] for all samples, resulting in 20 factors (Supplementary Data 1). These factors were inspected for variance explained, modality contribution, and correlation with cell type density scores[13,47] (Fig. 3b). While Factor 1 had the highest overall variance explained, the contribution was predominantly driven by the MSI modality. Several other factors, e.g., Factors 2, 4, 8, 16, and 19, displayed contributions from both modalities. Factor 2 showed a strong correlation with many disease-relevant cell types, including fibroblasts, myofibroblasts, interstitial/recruited macrophages, and other immune cell populations. Factors 8, 16, and 19 overlapped with alveolar type II (AT2) cells, while Factor 4 was associated with club, goblet, and ciliated cells commonly present in the ciliated epithelium of the lung (Fig. 3b, and Supplementary Fig. 2a).

Upon spatial inspection, we noted an overlap of areas with high Factor 2 activity and those manually annotated as suspected fibrosis (Fig. 3c, and Supplementary Fig. 2b). Examination of genes and peaks with the highest contribution to this factor revealed a distinct molecular signature which included a mix of genes associated with extracellular matrix (ECM) components (e.g., *Mfap4, Col4a1, Col4a2, Fn1, Bgn, Eln*), macrophage activity (e.g., *Cd74, Apoe, Psap*), immune functions (e.g., *H2-Ab1, H2-Eb1, H2-Aa, H2-D1*), and cathepsins (e.g., *Ctsh, Ctsd, Ctsb, Ctss*) (Fig. 3d). Several of these genes, such as *Psap, Cd74, Cd9*, and multiple cathepsins, align with markers previously described for a damage-associated macrophage population implicated in lung injury and fibrosis, where they play a role by modulating tissue responses to lung injury[49] and suggests immune-modulatory and matrix remodelling roles within the fibrotic regions. These findings were consistent with many of the fibrosis-associated genes detected in our previous Visium data analysis[13], however, a new dimension of metabolomic information could now be directly associated with the transcriptomic alterations.

The top mass spectrometry peaks contributing to Factor 2 were manually annotated as lactate (m/z 89.02435) and pantothenate (m/z 218.10372), followed by several long-chain fatty acids (FA) (e.g., m/z 253.21621, FA 16:1 palmitoleic acid; m/z 277.21621, FA 18:3 linolenic acid; m/z 305.24751, FA 20:3 eicosatrienoic acid; and m/z 303.2324, FA 20:4 arachidonic acid) and amino acids (e.g., m/z 118.05092, threonine; m/z 146.04588, glutamate; m/z 145.06187, glutamine; and m/z 154.06207, histidine) (Fig. 3d). Lactate has been repeatedly associated with pulmonary fibrosis, signifying a metabolic shift to anaerobic glycolysis, and may be directly involved in TGF-β-induced myofibroblast differentiation[46,50–52]. Factor 2 was further marked by metabolites of the citric acid cycle (TCA cycle; m/z 173.0092 aconitate) and glutamine metabolism, which have been implicated in TGF-β-induced myofibroblast activity[53]. The detection of TGF-β-related mechanisms within the fibrotic niche is also supported at a transcriptomic level, where fibrosis-associated TGF-β-signalling was previously reported based on the spatial gene expression data from IPF and BLM-injured lungs[13].

As the identified genes and metabolomic peaks associated with Factor 2 and regions of fibrosis remained somewhat contained within their separate modalities for this analysis, we aimed to better deduce the connectivity and semantic links between these genes and metabolites. By constructing a covariation network using GENIE3[54], a gene regulatory network comprising both genes and metabolite peaks could be inferred (Fig. 3e, and Supplementary Data 2). Many genes were interconnected based on their similar functions, such as those encoding antigen presentation molecules, ECM-related proteins, as well as genes highly expressed by macrophages. Threonine and lactate metabolites emerged as hub nodes, pointing towards the involvement of their metabolism across a wider range of biological processes.

Among them, threonine and lactate displayed covariation with *Apoe, Psap*, and *Fn1*, aligning with a previously identified fibrotic macrophage signature[49]. Additionally, lactate has been shown to induce profibrotic genes in macrophages[55], a finding supported by their spatial covariation in our analysis. Another coregulated group of genes and peaks in the network was annotated to molecules involved in prostaglandin (m/z 351.21693, m/z 333.2071) and hydroxyretinoic acid (m/z 315.19698) metabolism, which associates with the genes *Lrg1, Cd63, Clu*, and *Fn1*. The presence of eicosanoids, including various prostaglandins, has previously been reported to be elevated in BLM-induced pulmonary fibrosis[46] and of relevance for fibrotic progression[56], making it an intriguing subject for further studies given its co-localisation with an increased gene expression of ECM constituents.

In summary, our MAGPIE-enabled integrated spatial multi-modal analysis of the BLM-injured mouse lungs both complemented the histopathological assessment of the tissue and further revealed, to our knowledge, new links between transcriptomic and metabolic alterations occurring during lung fibrosis.

## MAGPIE facilitates spatial co-detection of drug substance and drug-induced transcriptional responses

The potential for spatial multi-modal analyses extends to many areas of application and, as such, we tested its unique utility in studying transcriptional responses linked to local drug distributions within a tissue. Through Visium and DESI MSI spatial multi-omics data collection on consecutive sections, and subsequent co-registration using MAGPIE, we could detect the localisation of compound AZX in a rat lung sample following inhaled administration, and explore its impact on the surrounding tissue (Fig. 4a). The inhaled administration of AZX, a drug previously in preclinical development by AstraZeneca, has been associated with adverse histopathological lung lesions in animal toxicology studies, thus prompting further exploration of its lung distribution and assessment of unwanted effects in situ.

Visual inspection of the AZX-dosed tissue revealed that dense AZX hotspots, as detected by MSI, co-localised with variably shaped luminal crystal-like structures observed on microscopic examination of the Visium H&E images (Supplementary Fig. 3). After aligning the transcriptome and metabolome data into a shared coordinate space using MAGPIE, we investigated the spatial relationship between AZX tissue depositions and local gene expression patterns. The Visium data was first deconvolved into 20 transcriptomics-driven factors using non-negative matrix factorisation (NMF). Next, the spatial cross-correlation between the transcriptomic NMF factors and the intensity of the AZX compound (m/z 550.15724), as captured by the MSI modality, could be computed directly owing to the integrated coordinate system. The results identified several transcriptomic NMF factors which showed a spatial association with the compound hotspots, of which Factor 14 showed the highest spatial cross-correlation with the compound, followed by Factors 8 and 4 (Fig. 4b). Of these three factors, Factor 14 displayed the most distinct spatially cohesive pattern that corresponded well to the distribution of AZX (Fig. 4c) and thus supported a transcriptomic signature localised to areas of drug deposition. Investigation of the underlying Factor 14 transcriptomic signature revealed a profile enriched for immune and defence response pathways, particularly in response to external and chemical stimuli (Fig. 4d). This transcriptional signature, identified through the use of integrated Visium and MSI data, suggests a localised inflammatory molecular response triggered by the drug deposition. It further proposes a unique strategy to assess mechanisms of drug-associated toxicity, aiding the development of compounds with a more favourable safety profile.

Here, we illustrated that multi-modal analyses of compound-exposed tissues can provide insights into drug distribution and its associated effects within the tissue and that MAGPIE offers a

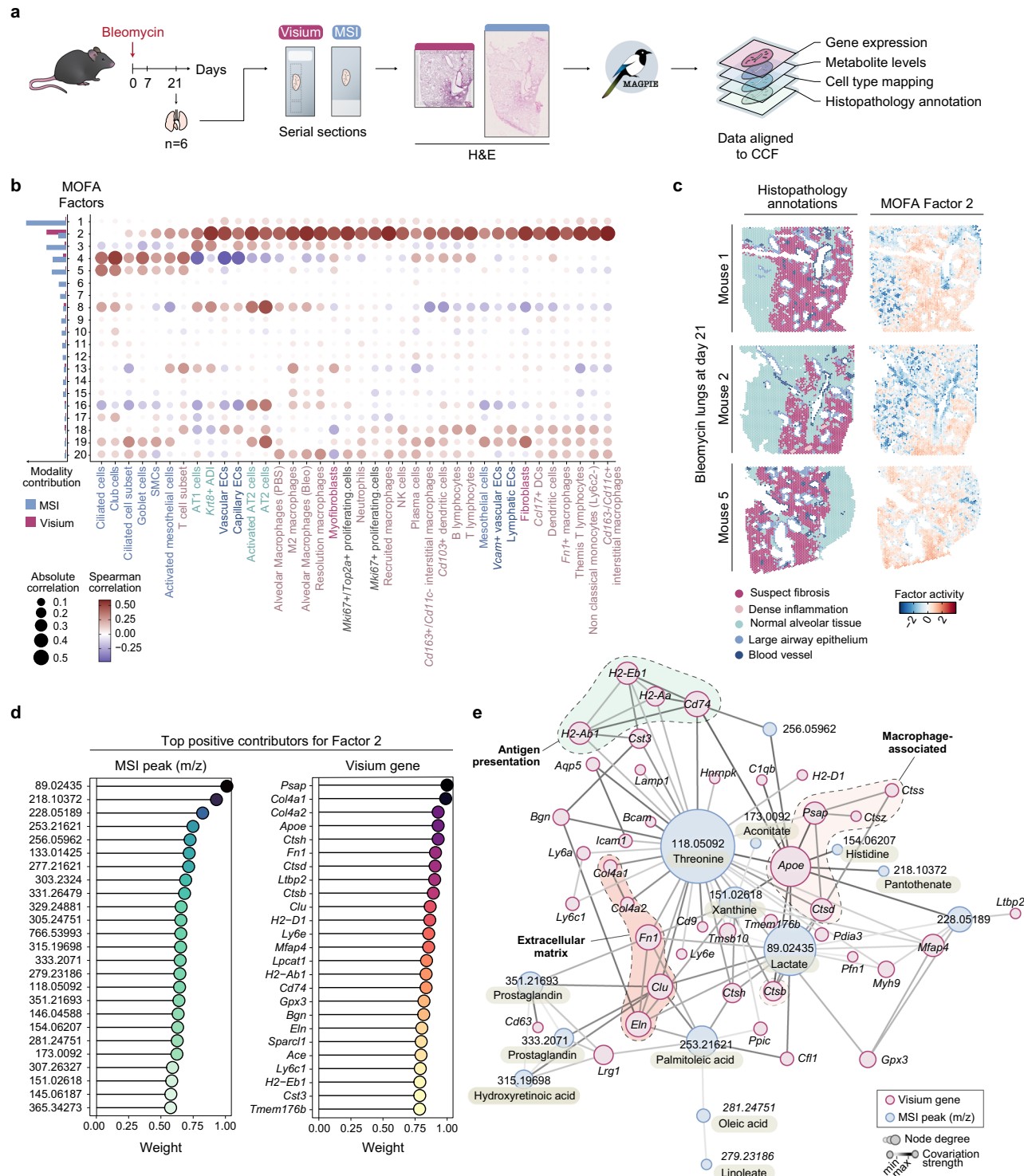

**Fig. 3 | Integrative spatial transcriptomic and metabolomic analysis of bleomycin (BLM)-induced lung fibrosis. a** Experimental and computational workflow for generation of spatially resolved transcriptomic (Visium; previously generated[13] and publicly available data under accession GSE267904) and metabolomic (DESI MSI) datasets from a mouse model of acute BLM-induced pulmonary fibrosis, where lung samples were collected at 21-days following BLM administration. **b** Multi-omics factor analysis (MOFA)[29,48] results on aligned Visium and MSI data with 20 factors. Cell type mapping was previously performed using annotated scRNA-seq dataset published by Strunz et al.[47]; inferred cell type densities were correlated with the MOFA factor activities across all spatial locations using Spearman correlation, shown through the colour of the points. The relative contribution of modalities for each factor is summarised as a bar chart on the left side, with colours reflecting the modality. **c** Spatial plots of the histopathology annotations,

based on the Visium H&E images, next to the activity of MOFA Factor 2 for three sections, both shown through colour. Similar analysis on 3 further samples is shown in Supplementary Fig. 2b. **d** Weight of the positive contribution of peaks and genes to Factor 2, showing the top 25 features. The colour of points reflects the relative weight. **e** Multi-omics covariation network analysis on top 50 genes and top 25 peaks in Factor 2. The graph was constructed using GENIE3[54] where the top gene-gene, gene-peak and peak-peak edges are selected for the network visualisation. Edge weights (darker edge colour) are proportional to covariation strength from GENIE3 and node sizes are proportional to node degree. Nodes, representing genes and peaks, are coloured by modality. CCF common coordinate framework, SMCs smooth muscle cells, ADI alveolar differentiation intermediate, ECs endothelial cells, DCs dendritic cells. Source data are provided as a Source Data file.

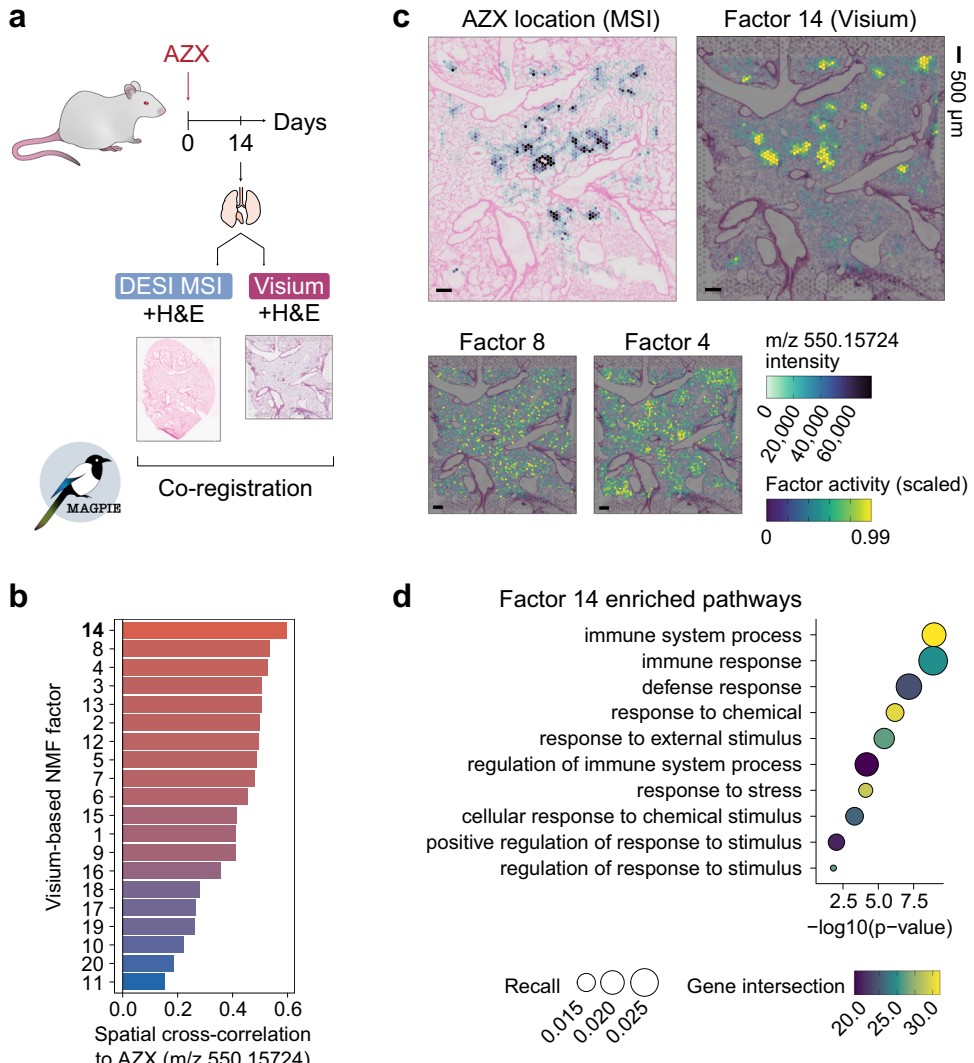

**Fig. 4 | MAGPIE co-registration of spatial multi-omics datasets from AZX-treated rat lung. a** Outline of experimental design and Visium and DESI MSI data generation from a rat lung exposed to the AZX compound. **b** Spatial cross-correlation assessment of overlap between MSI AZX compound distribution (m/z 550.15724) and Visium NMF factors. The colour of bars reflects the spatial cross-correlation. **c** Spatial distribution of MSI AZX intensity distribution (capped at 99th percentile) and the scaled (0-1) Visium NMF factor activities for Factors 14, 8, and 4 (also capped at 99th percentile), shown through colour overlayed on H&E images.

Only one sample was analysed for illustration. Scale bars reflect 500 μm. **d** Enrichment analysis on the top 100 contributing genes for NMF Factor 14, showing the Gene Ontology (GO) enriched terms. Enrichment analysis was performed using a rank-based cumulative enrichment test as implemented in g:profiler[75] with g:SCS (Set Counts and Sizes) multiple testing correction. Colours represent gene intersection size between queried genes and genes annotated to each term; dot size corresponds to the recall (ratio between gene intersection size and number of term genes). Source data are provided as a Source Data file.

streamlined approach to correlate pharmacological compound localisation and metabolomic signatures with molecular responses at tissue resolution.

**Spatial multi-modal analysis of human breast cancer through MAGPIE reveals changes in tumour microenvironment**
The characterisation of tumour microenvironments is another research topic benefiting from spatial multi-omics analysis facilitated by the MAGPIE framework; here, we applied the workflow to three invasive lobular breast cancer samples from Godfrey et al.[57] (Fig. 5a).

After co-registering and aggregating MSI pixels to Visium spots, we performed MSI-driven k-means clustering (k = 2) and assigned labels (cancer and non-cancer) to each cluster using H&E images (Fig. 5b). Cell type deconvolution of Visium data using CARD[58] with a single-cell transcriptomic breast cancer dataset[59] confirmed the localisation of cancer epithelial cells to our metabolically-assessed 'cancer'

regions (Fig. 5c), of endothelial cells to 'non-cancer' regions, and of T-cell clusters around the periphery of cancer regions (Supplementary Fig. 4a).

To further illustrate the localisation of cell types, genes and metabolites, we calculated the radial distances from cancer regions (Fig. 5d), and analysed the relationship between radial distance and inferred cell type densities (Supplementary Fig. 4b). This allowed us to characterise the Visium-derived cellular composition in relation to MSI-derived cancer regions, revealing that cancer epithelial cells showed a sharp decrease in density with increasing distance from cancer regions, while plasmablasts, representing active immune infiltration, declined more gradually. At the tumour boundary, we observed enrichment of cancer-associated fibroblasts (CAFs) consistent with their role in tumour invasion through extracellular matrix remodelling.

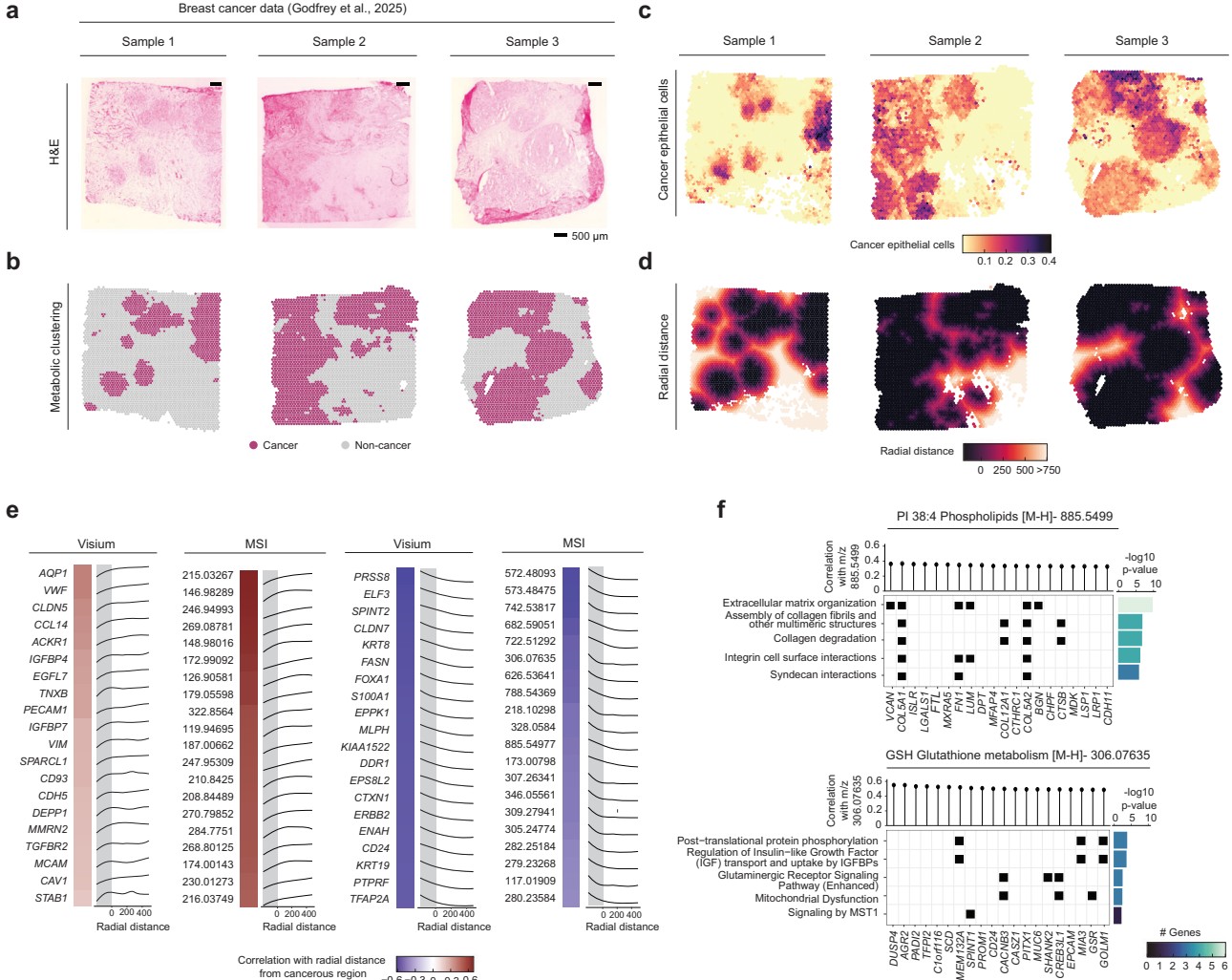

**Fig. 5 | Analysis of tumour microenvironment in human breast cancer through MAGPIE. a** H&E stained tissue sections for three breast cancer samples from Godfrey et al.[57]. Scale bars reflect 500 μm. **b** Metabolic clustering (k-means, k = 2 per sample) into cancer and non-cancer regions using MSI data shown spatially through colour. **c** Spatial distribution of cancer epithelial cells cell type deconvolution scores (using CARD[58] with single-cell reference from Wu et al.[59]) shown through colour. **d** Radial distance calculated from metabolic clustering cancer regions shown spatially through colour. **e** Top 20 positive (left) and negative (right) genes and metabolite peaks correlated with radial distance, up to 500 μm. The Spearman correlation is shown in the colour of tiles on the left of each panel and line plots

reflecting smoothed expression/intensity across radial distance are shown on the right. **f** Top genes correlated with selected peaks, m/z 885.5499 PI 38:4 and m/z 306.07635 GSH. Gene set over-representation analysis was performed on the top 20 genes in each case using Ingenuity Pathway Analysis (IPA)'s right-tailed one-sided Fisher exact test and the top five terms are shown, indicating the genes associated with each term. At the top, the Spearman correlation of each gene with the focus metabolite peak is shown, and on the right is a bar plot where the x-axis shows -log_{10}(p-value) of the enriched term and the colour reflects the number of top correlated genes shown associated with each term. Source data are provided as a Source Data file.

We then correlated radial distance with individual genes and metabolite peaks to find molecules localised to specific regions (Fig. 5e), including genes high in tumour core corresponding to known epithelial (luminal) tumour markers (e.g., *KRT8, KRT19, FOXA1*), to RTK-linked signalling and motility (e.g., *ERBB2, DDR1, ENAH*), and to metabolic reprogramming and stress responses (e.g., *FASN, S100A1*). Specific metabolites were identified as preferentially localised in the tumour core, such as pantothenic acid (vitamin B5, *m/z 218.10298*), which is associated with high MYC expression in human mammary tumours[60]. Ceramide species, which are closely related to the necrotic nature of the tumour core, were detected within this niche, particularly Cer(d34:1) (*m/z 572.48093*), which is a known marker of necrotic tissue reported as absent from viable tumour regions[61].

Further, we detected several genes localised to both the immediate tumour microenvironment and more distal tissue regions, such as genes involved in extracellular matrix and stromal

remodelling, including *TNXB, SPARCL1* and *STAB1*, genes regulating immune cell recruitment and modulation within the tumour microenvironment. e.g., *CCL14, ACKR1, TGFBR2*, and markers representing the more structured vasculature outside the tumour core, e.g., *PECAM1, VWF, CDH5, CLDN5, CD93*. The metabolic profile of the distal region was predominantly characterised by glucose (*m/z 215.03267*), the top positively correlated peak, in agreement with reports of pronounced glucose depletion observed in the tumour core relative to the surrounding stroma[62].

Next, we looked for the strongest colocalisation between genes and selected metabolites within the tumour microenvironment (Fig. 5f). This approach uncovered gene-metabolite associations consistent with known tumour biology. For example, phosphatidylinositol (PI (38:4), *m/z 885.5499*) was strongly associated with extracellular matrix (ECM)-related genes, including *COL5A2, COL5A1*, and *FN1*. PI and its phosphorylated derivatives (phosphoinositides, PIPs) are known to

play a crucial role in cancer development and progression, particularly through remodelling of the ECM in the tumour microenvironment[63]. Moreover, glutathione (GSH, *m/z 306.07635*) strongly correlated with genes such as *GSR*, *CREB3L*, and *CACNB3*, all of which are involved in mitochondrial dysfunction, a relationship supported by previous studies[64]. In addition to elucidating established biological processes, MAGPIE uncovered, to the best of our knowledge, novel insights and associations between metabolites and genes. For example, a strong correlation was observed between GSH and several genes, including *GOLM1, MIA3*, and *TMEM132A*, which are implicated in post−translational protein phosphorylation, interactions which, to our knowledge, have not been previously described.

Through this re-analysis of the human breast cancer data, we illustrated how MAGPIE can resolve the cellular and molecular heterogeneity and complexity of tumours by capturing spatial relationships between cell types, metabolic gradients, and gene expression patterns across tumour cores, stromal interfaces, and surrounding tissue. Together with our lung fibrosis and drug deposition studies, this demonstrates MAGPIE's utility across a range of complex diseases, to provide molecular-level mechanistic understanding within intact tissues but also to provide the basis for new testable biological hypotheses.

## Discussion

Through the MAGPIE computational workflow, supported by an optimised lung tissue sampling strategy, we have demonstrated the impactful value of integrating spatial transcriptomic and metabolomic datasets to extract insights into disease mechanisms and drug-induced injury in tissues. The MAGPIE computational workflow showed versatility across different species and tissue types and has been successfully applied to both consecutive section and same-section profiling in human, rat, and mouse datasets. The pipeline was also applied to datasets using both MALDI and DESI mass spectrometry imaging, using varied ionisation modes and matrix choices, further discussed in Supplementary note 4. We envision MAGPIE to be of high value for users seeking an accessible and generalisable pipeline for integrating spatial multi-modal data.

MAGPIE allows users the flexibility to incorporate external information and optimise hyperparameters to suit their available samples. Based on our input parameter tests and outcomes, we offer several recommendations for future datasets processed with MAGPIE for spatial multi-modal integration, particularly when dealing with serial sections. Obtaining a microscopy image from the MSI samples, preferably using the same H&E staining technique as for Visium, is important for accurate landmark identification between MSI and Visium images. High structural similarity between the matching tissue sections should be prioritised whenever possible, especially when consecutive or more distant sections are used for the two modalities. Our examples highlight the need for careful consideration of feasibility before attempting co-registration and that accurate co-registration may require a greater number (≥10) of landmarks and/or a non-linear transformation, depending on the similarity between sections. We advise careful assessment of section similarity and a comparison of different co-registration options when producing new datasets.

Moreover, we illustrate the unique benefit of obtaining and integrating paired spatial multi-omics data for studies of disease characteristics as well as local tissue responses to drug depositions. In the BLM mouse model, the integration of spatial transcriptomics and metabolomics revealed distinct signatures associated with fibrotic regions. These alterations were characterised by an increased transcriptionally-inferred presence of inflammatory cells, fibroblasts, and myofibroblasts, coupled with a shift in local metabolism towards states previously described to be relevant for pulmonary fibrosis[46,51,52]. Additionally, the integration of Visium data with MSI-derived information on compound deposition in the lung using MAGPIE enabled

the identification of an inflammatory transcriptomic signature localised to areas of drug accumulation in the rat lung, potentially providing mechanistic insight into the observed lung toxicity of this compound. We further highlighted MAGPIE's utility in the cancer setting and unpicked spatial relationships between cell types, metabolic gradients and gene expression patterns across tumour cores, stromal interfaces, and surrounding tissue.

Although the MAGPIE pipeline is designed to utilise the standard file format for spatial data introduced by Visium, the input format of the second modality is a simple tabular matrix containing features and spatial coordinates. Therefore, it may be further applicable to measurements from other omics beyond MSI as more combinations of modalities become possible. For example, as co-registration with images from the same tissue section is already supported by the MAGPIE pipeline, it is readily applicable to combined assessment of metabolomics, transcriptomics, and proteomic data, e.g., from multiplex immunofluorescence imaging. The co-registration of spatial transcriptomics with spatially resolved proteomics or spatially resolved chromatin accessibility profiling[65,66] would enable a greater understanding of the dynamics of gene regulatory relationships in space. This could be achieved through inference of gene regulatory networks using computationally combined observations across modalities, by adapting single-cell methods[67–69] or using novel spatially resolved methods as the analysis landscape continues to develop.

One inherent limitation of MAGPIE derives from the use of data inputs derived from fresh frozen tissues. Histology provides essential structural context for spatial omics analyses, as demonstrated through our case studies, and formalin-fixed paraffin-embedded (FFPE) tissue samples offer an attractive alternative to fresh frozen tissue for pathological interpretation due to the prevalence of freezing artefacts. While workflows for Visium and other SRT methodologies have been established for FFPE tissues, integrating MSI with FFPE samples requires careful consideration of sample preparation protocols (Supplementary note 4). As these protocols advance, MAGPIE's flexible design will allow integration of multi-modal datasets from both FFPE and fresh frozen tissues. Alongside these technical considerations, we acknowledge there are also some intrinsic biological and analytical limitations to current spatial multi-omics approaches, including the static nature of single-timepoint tissue sections, limiting the possibility for causal inference between transcriptomic and metabolomic changes, and the multicellular resolution of platforms like Visium and MSI which can obscure single-cell-level heterogeneity (Supplementary note 4). Another limitation is the mismatch is spatial resolutions between Visium and MSI dataset, where Visium captures 55 μm spots while MSI pixel sizes vary by technology and acquisition settings and can exhibit signal diffusion. Users should therefore interpret co-localisation findings with attention to the resolution disparity in their dataset, assisted by MAGPIE's flexible aggregation strategies designed to mitigate resolution differences and false-negative results.

In conclusion, we present MAGPIE as a robust, versatile, and reproducible pipeline for the co-registration and computational integration of spatial transcriptomic and metabolomic data, which enables richer biological insights from tissues, showcased by its application to characterise molecular responses to tissue injury. By continuing to refine and expand spatially resolved analysis techniques, and in parallel developing new computational approaches for processing and analysing the data, we can further enhance the resolution and applicability of spatial multi-omics analyses in biomedical research and enable the generation of previously unattainable molecular insights into human disease for the discovery of novel treatments.

## Methods

### Experimental data generation

**Ethical considerations.** Animal care and handling adhered to the standards established by the Council of Europe ETS123 AppA, the

Helsinki Convention for the Use and Care of Animals, Swedish legislation, and AstraZeneca global internal standards. All experiments were ethically approved by the Gothenburg Ethics Committee for Experimental Animals in Sweden, complying with Directive 2010/63/EU. The studies received local Ethical committee approval in Gothenburg (EA000680-2017 and 2020-002853) with the assigned site number 31-5373/11.

### Animals

**Bleomycin-induced lung fibrosis mouse model.** The mouse lung sample generation and collection were previously described in the study by Franzén and Olsson Lindvall et al.[13]. In short, female C57BL/6NCrl mice (Charles River, Germany; eight weeks of age upon arrival) were housed in a facility with ad libitum access to food (R70, Lantmännen AB, Vadstena, Sweden) and tap water. Following a five-day acclimatization period, mice were subjected to oropharyngeal administration of either 30 µl of bleomycin (Apollo Scientific, BI3543, Chemtronica Sweden; 40 µg/mouse) dissolved in saline or saline alone (vehicle control). Lung samples were collected on day 7 (d7) and day 21 (d21) post-bleomycin challenge, capturing both the early inflammatory and tissue remodelling phase (d7) and established tissue damage (d21).

**AZX-dosed rat lung.** Male Wistar Han rats (Charles River, Germany) were seven weeks old upon arrival and ~10 weeks at the start of dosing. They were group caged, 2–6 rats per cage, and maintained on a normal day/night cycle of dark 18:00 and light 06:30 at a uniform temperature ($>25\,^{\circ}$C) and humidity ($<5\%$), with access to chew sticks and nesting material. Rats had free access to diet (R70) and tap water. Animals were randomised and regrouped into new cages prior to the first dose and housed with others in the same dosing group to minimise contamination by grooming and coprophagia. The animal used in this study ($n=1$) was subjected to daily inhalation exposure via snout-only administration for 14 days with 15 mg/kg/day of the AZX test formulation consisting of AZX (10%), MPEG-2000-DSPE (1.25%), trileucine (5%), and trehalose (83.75%), in a flowpast exposure chamber. The animal was acclimated to the method of restraint over a five-day period preceding the first test substance exposure. Animal body weight was recorded prior to the study, twice weekly, pre-dose, and at termination.

**Tissue processing for spatial multi-modal analysis of rodent lung.** The rodent lung tissue sampling protocols were optimised to enable high quality analysis using both the Visium Spatial Gene Expression platform and mass-spectrometry imaging (MSI), both with regards to analyte preservation and histological integrity. Fresh frozen tissue specimens are typically embedded in optimal cutting temperature (OCT) compound for Visium analysis, however, OCT is known to interfere with the quality of mass spectrometry readouts and should therefore be avoided or removed in MSI experiments[21,22]. Thus, we opted for careful inflation of the lungs using a low-melting point agarose solution prior to snap freezing of the tissue without any further embedding material. We employed agarose inflation in order to preserve anatomical structure and maintain natural spatial relationships between tissue compartments (Supplementary note 1). This approach is routinely used in lung histology and is critical for preventing collapse of the delicate alveolar architecture. However, MAGPIE is not reliant on agarose inflation and can be applied to tissues prepared via standard protocols appropriate for other organs.

For the mouse study, the mice were first anaesthetised using isoflurane (5% concentration, air flow ~2 L/min) and maintained with 3% isoflurane (air flow ~0.7 L/min). A midline incision was made from the abdominal midsection to the chin. Subsequently, 0.1 mL of heparin was injected through the diaphragm into the heart, followed by severing the abdominal aorta to exsanguinate the mice. The heart and right lung lobes were tied off and removed. The pulmonary circulation was perfused with 37 °C saline followed by 37 °C low-temperature melt agarose (SeaPlaque) solution. The lung was then gently inflated with 0.4-0.5 mL of 37 °C agarose solution via the trachea and tied off. The lung tissue was collected and snap-frozen in pre-chilled NaCl over dry ice and stored at -80 °C for further analyses.

For the rat sample, the trachea was surgically exposed and secured using a 2−0 silk suture on the day of termination. Subsequently, 0.4 mL of heparin was injected into the heart through a small incision in the diaphragm, after exposing the abdominal wall. The pulmonary circulation was flushed via the pulmonary artery using a solution of 15 mL of 37 °C saline followed by 7 mL of 37 °C low-melting point agarose (SeaPlaque) solution (0.75 g agarose in 50 ml saline). A suture was placed around the heart to prevent heart leakage. A cannula was then introduced into the trachea, and the lung was gently inflated with 4.5 mL of agarose solution. Lungs and heart were removed as a single unit, placed in ice-cold PBS, and allowed to rest on wet ice for ~20 minutes to solidify the agarose. The right superior lobe of the lung tissue was thereafter placed on pre-chilled tin foil in a plastic fixation cassette. The sample was rapidly frozen by immersion in isopentane, pre-chilled in liquid nitrogen. Samples were stored at -80 °C until further analysis.

### Spatial transcriptomics and metabolomics experimental procedures

**Sectioning and RNA assessment.** Agarose-inflated mouse and rat lung tissues were mounted to the cryostat specimen chuck using a drop of purified water, and thereafter cryo-sectioned at 10 µm thickness with the cryostat chamber temperature set to -20 °C and -10 °C for the specimen head. It is important to note that the frozen tissue samples are never embedded in OCT compound, given its incompatibility with MSI analysis. Consecutive sections from each lung were thaw-mounted on a Visium slide for spatial transcriptomics and a Superfrost slide (Fisher Scientific, Loughborough, UK) for desorption electrospray ionisation (DESI) MSI. Slides were stored in -80 °C until further analyses. For RNA quality assessment, ~ten sections from each lung tissue were collected and stored at -80 °C prior to RNA extraction using the RNeasy micro kit (Qiagen). RNA quality was assessed using a 5300 Fragment Analyzer (Agilent), and the RIN values were > 8 for all samples.

**Generation of spatially resolved transcriptomics data.** For the AZX-treated rat sample, tissue fixation and staining followed the Methanol Fixation, H&E Staining, and Imaging Visium protocol (10X Genomics). Imaging was performed at 40X magnification using the Aperio Digital Pathology Slide Scanner (Leica Biosystems), and sequencing libraries were prepared according to the Visium Spatial Gene Expression User Guide (10X Genomics; Rev E). The rat library ($n=1$) was sequenced on the NovaSeq 6000 (Illumina) platform with an S4 flowcell using the following set-up: Read1: 28 bp, Index 1: 10 bp, Index 2: 10 bp, Read2: 90 bp. A 1% PhiX spike-in was included in each run. A total of 177 M reads was generated for the rat sample.

**Desorption electrospray ionisation mass spectrometry imaging (DESI MSI).** For both mouse ($n=11$) and rat ($n=1$) lung samples, DESI MSI was carried out using an automated 2D DESI source (Prosolia Inc, Indianapolis, IN, USA) with a home-built sprayer assembly mounted to a Q-Exactive FTMS instrument (Thermo Scientific, Bremen, Germany). Analyzes were performed at spatial resolutions of 65 µm in negative ion mode and mass spectra were collected in the mass range of 80 − 900 Da with mass resolving power set to 70000 at $m/z$ 200 and an S-Lens setting of 100. Methanol/water (95:5 v/v) was used as the electrospray solvent at a flow rate of 1.0 µL/min and a spray voltage of -4.5 kV. Distance between DESI sprayer to MS inlet was 7 mm, while distance between sprayer tip to sample surface was 1.5 mm at an angle of 75°. Nitrogen N4.8 was used as nebulising gas at a pressure of

6.5 bar. Omnispray 2D (Prosolia, Indianapolis, USA) and Xcalibur (Thermo Fisher Scientific Inc) software were used for MS data acquisition. Individual line scans were converted into centroided .mzML format using MSConvert (ProteoWizard toolbox version 3.0.4043) and subsequently into .imzML using imzML converter v1.3. Haematoxylin and eosin (H&E) staining was performed post-analysis on the same tissue sections and the stained sections were imaged at 20x with Aperio CS2 digital pathology scanner (Aperio Tech., Oxford, UK), and visualised with QuPath 0.23[70] for histopathological annotations performed by a pathologist (L.S.).

### Computational data processing

**Preprocessing of mass spectrometry data.** For the bleomycin mouse model, AZX-treated rat and human breast cancer datasets, the .imzml data files were initially imported into SCiLS Lab software (Bruker Daltonics, Germany, 2022a MVS). Bisecting k-means segmentation was performed within SCiLS Lab to create two regions of interest (ROIs): tissue and background (area outside the tissue). Individual samples were then segmented using the tissue/background ROIs with some manual alterations. Individual pixel-level total ion count (TIC) normalised peak intensities were extracted using the SCiLS Lab API, which yields a pixel-by-peak table with associated metadata. This table was later used as input to the MAGPIE pipeline. For the bleomycin mouse model and AZX-treated rat, peaks were annotated against compounds in the KEGG database[71] using [M-H]⁻ adducts with a mass tolerance of 5 ppm. Only peaks which could be annotated to a metabolite were retained for the downstream analysis of the bleomycin-treated mouse samples.

**Preprocessing of Visium data.** Raw FASTQ files were processed with Space Ranger 1.2.2 for mouse bleomycin samples and 1.3.1 for rat AZX sample (10x Genomics). Sequencing reads were aligned to their respective reference genomes: rn6 for the rat sample and mm10 for the mouse samples. Alignment of H&E images to the fiducial frame was performed manually using the Loupe Browser software (v.6, 10X Genomics). For the bleomycin-treated mouse study, the Space Ranger output data and associated metadata, including cell type deconvolution results, was published in conjunction with the publication by Franzén and Olsson Lindvall et al.[13], and can be downloaded from Gene Expression Omnibus (GEO) (accession number GSE267904) and BioStudies (S-BSST1409).

**The MAGPIE pipeline.** The MAGPIE pipeline is written in Python (v3.10.11) and presented as a Snakemake workflow, with dependencies on snakemake, shiny, matplotlib, pandas, numpy, scikit-image, pathlib, scikit-learn, scipy, json, collections, shutil, gzip, h5py, and scanpy. All pipeline components can be initiated from the command line; detailed documentation for each component is provided at (https://core-bioinformatics.github.io/magpie). The full pipeline is open source under the MIT/ CC-BY licence and is found on GitHub at (https://github.com/Core-Bioinformatics/magpie).

As input, the MAGPIE pipeline takes the Visium data in the standard Space Ranger output format (including *filtered_feature_bc_matrix.h5* containing gene expression information, *tissue_hires_image.png* with the H&E image, *tissue_positions.csv* containing spatial coordinates per spot and *scalefactors_json.json* translating from H&E image to spatial coordinates) and a peak-by-pixel table for the MSI data, with the option to add a microscopy image for the MSI section. The image alignment within MAGPIE utilises a landmark-based approach. Users may opt to select their own landmarks outside the pipeline, either manually or automatically using an external tool (e.g., ELD[23]). MAGPIE includes an interactive Python Shiny application which can be used before the Snakemake pipeline. Within this application, a lower dimensionality image is created based on the MSI data using a user-selected method, for example, using the first or first 3

principal components (based on all or a subset of peaks) as colour channels or using an individual peak of interest based on a priori knowledge. For manual landmark identification, users are thereafter prompted to select landmarks within the interactive application, either (1) between MSI dimensionality reduction image and Visium H&E image or (2) between MSI dimensionality reduction image and MSI microscopy image, if available, then between the MSI microscopy image and Visium H&E image. If users identify landmarks externally, they may add them in a tabular format and skip the in-built landmark selection tool.

The next stage of the pipeline is streamlined within a Snakemake workflow, which consists of three main modules. In the first module, the identified landmarks are used to map MSI coordinates to Visium coordinates, using either a linear (affine) or non-linear (TPS) transform based on the scikitimage implementation (v0.24) in accordance with user selection. This results in a common coordinate framework (CCF) between the two modalities where each (x,y) coordinate in the MSI modality can be directly mapped into the Visium modality. The second module of MAGPIE prepares and stores the new MSI data in a Space Ranger-style object, including a .h5 file and a 'spatial' folder containing coordinates and images, which can be read by other spatial analysis ecosystems. As a third optional module, the data points for the MSI modality can be aggregated into the Visium spot barcodes, creating a 1:1 spatial map between the MSI and Visium observations, following a strategy described in the next section.

**Aggregation of MSI data per Visium spot.** For performing fully integrated spatial multi-modal analysis, we match MSI pixels to Visium spots to create combined 'observations'. Initially, we calculate the between-spot distance in the Visium data then find the MSI pixels within half that distance for each Visium spot (the spots' expanded radius) (Fig. 1c). This is calculated efficiently by finding the k-nearest neighbours for each spot and then selecting the neighbours within the specified expanded radius distance. MSI pixels within each expanded Visium spot are then either averaged (mean or weighted by distance) or summed according to user choice to create the new matching observations. Alternatively, users can select to use the actual Visium spot radius, for example in cases where MSI data is of much higher spatial resolution that Visium and so MSI pixels may fall wholly outside Visium spots, or use the estimated MSI pixel size in cases where MSI data is of lower resolution to Visium and so an MSI pixel may contribute to multiple Visium spots (see Supplementary note 2). This step of the data integration was implemented within the MAGPIE Snakemake workflow as a final optional module and as a separate function, CreateMultiModalObject, in the semla R package (v. ≥ 1.3.0)[26], with which the loaded Visium and MAGPIE-aligned MSI data can be joined into a multi-modal object.

### MAGPIE pipeline benchmarking

**Hyperparameter testing.** The performance of co-registration was assessed on the overlap between MSI and Visium observations after coordinate transformation. We relied on the tissue/background annotation identified through the Space Ranger pipeline and applied tissue/background labels to the MSI data based on the first principal component, using data-specific thresholds identified using underlying microscopy and lower dimensional images. We then calculated summary statistics to capture the accuracy of the transformation to match tissue to tissue and background to background (i.e., # spots annotated as tissue in both modalities + # spots annotated as background in both modalities divided by total # spots). For each sample (n = 8) in the bleomycin-treated mouse dataset, 20 landmarks were identified (1) between MSI dimensionality reduction and MSI H&E then between MSI H&E and Visium H&E and (2) between MSI dimensionality reduction and Visium H&E directly. For numbers of landmarks ranging between 4 and 20 (with 5 repeats), the given number of landmarks was sampled

from the total pool, the transformed coordinates calculated and pooled into matching Visium/MSI spots and the alignment accuracy of tissue/background labelling recorded. Samples were ranked for image similarity based on the best accuracy achieved across all numbers of landmarks and repeats.

**Same-section spatial multi-modal data.** For the same-section Visium MALDI MSI multi-modal data tests, the datasets were acquired from Mendeley Data (DOI: 10.17632/w7nw4km7xd.1) and three mouse brain (V11L12-038_B1, V11L12-038_D1, V11T16-085_C1) and three human brain (V11T17-102_A1, V11T17-102_B1, V11T17-102_D1) datasets were selected. A total of 15 landmarks were identified per sample and a number of landmarks ranging between 3 and 15 (with 5 repeats) was tested using the same approach as described above.

### Downstream multi-modal data analysis

**Bleomycin mouse model of pulmonary fibrosis.** MAGPIE-aligned MSI data, using TPS transformation[72], for mouse lung samples collected 21 days after bleomycin administration (one tissue section per animal; $n = 6$) was loaded into R with semla v1.2.1. The corresponding Visium Space Ranger output data files (GSE267904[13]) were imported with ReadVisiumData, only keeping spots within tissue and image borders. The Visium data was filtered to exclude genes starting with Rpl, Rps, Mrp and mt-, pseudogenes and noncoding genes and focus solely on protein-coding genes detected in > 5 spots. Spots were further filtered to remove those with ≤100 transcripts and ≥10% mitochondrial or haemoglobin gene content. The MSI data was TIC-normalised beforehand, and once in a semla object format, it was filtered to keep pixels with more than 300 detected peaks and a total intensity of $> 14 \times 10^5$. Peaks that had not been annotated to metabolites (using M-H adduct) were excluded from the downstream analysis. Further, peaks with the same mass down to two decimal points were combined by taking the peak with the highest sum of intensities across all samples. A multi-modal semla object was thereafter created with mean aggregating function to combine m/z peak intensity values per Visium spot. The full Visium data was subsequently normalised and scaled with NormalizeData and ScaleData. All annotated MSI peaks (scaled per peak) and top 2000 most abundant genes were used to create a MOFA+[29] (v1.10.0) model with 20 factors (medium convergence mode). Cell type deconvolution results were obtained from previously generated data by Franzén and Olsson Lindvall et al.[13]; the Spearman correlation was calculated between inferred cell type densities and MOFA factor activities across spots. Factors showing the highest correlation with any cell type in the negative direction were reversed to favour interpretability (specifically factors 1, 2, 3, 4, 8, 9, 10, 12, 14, 15, 18, 19, 20). Spot factor scores were compared to histopathology annotations and the top genes and peaks in factors showing clear overlap with fibrotic and inflamed regions were inspected. Multi-omics covariation network analysis was performed on the top 50 genes and 25 peaks in Factor 2 using GENIE3[54] (v1.22.0). The Spearman correlation between all peaks and genes was also calculated and connections between nodes which showed negative correlation were removed to ensure all edges correspond to positive covariation. The top gene-gene, gene-peak, and peak-peak edges were selected (proportional to number of nodes included for each modality, specifically top 40 for gene-gene and gene-metabolite edges and top 20 for metabolite-metabolite) and the resulting network was visualised using the R package igraph (v2.0.3, Davidson-Harel layout algorithm, cooling factor 0.8). Disconnected subgraphs containing ≤ 3 nodes were removed from the network. Edge weights are proportional to covariation strength from GENIE3 and node sizes are proportional to degree in the network.

**AZX-dosed rat lung tissue.** After aligning observations between datasets, as detailed above, data from each modality was separately loaded into a semla[26] (v.1.2.1) object, removing spots outside of the tissue and image area. The Visium data was filtered to remove genes matching the regex patterns ^Rpl | ^Rps, for ribosomal genes and ^Mt-, for mitochondrial genes, genes annotated as non-coding or pseudogenes, as well as genes detected in 5 or fewer spots. Low-quality spots with ≤ 300 transcripts or ≥ 30% mitochondrial or haemoglobin-related gene content were further removed. The TPS-aligned and TIC-normalised MSI data was filtered to keep only peaks with a metabolite annotation and m/z 550.15724 (AZX compound), and pixels with more than 300 assigned peaks and a total (normalised) ion intensity measure exceeding $23 \times 10^5$ were kept. The joint multi-modal semla object was thereafter created using the function CreateMultiModalObject, with the data aggregation argument set to mean. Normalisation and scaling of the Visium data were thereafter performed using Seurat (v4.4.0) functions NormalizeData and ScaleData. NMF analysis, specifying 20 factors, was run on the Visium data using the RunNMF function from the singlet R package[73] v(0.0.99). Spatial co-localisation with m/z 550.15724 in the metabolomics modality was assessed using the spatial cross-correlation metric from MERINGUE[74] (v1.0). Enrichment analysis of Gene Ontology (GO) biological processes terms was performed on the 100 top contributing genes for NMF Factor 14 using the R package gprofiler2[75] (v0.2.1), with the organism set to rnorvegicus.

**Same-section spatial multi-modal mouse brain data.** Data from MS3 (V11T16-085_C1) was obtained as above and coregistered using MAGPIE (default parameters except agg_fn = 'weighted_average' and radius_to_use = 'msi' in create_barcode_matrix). Both modalities were read using semla[26] (v1.2.1). Visium data was filtered to remove low quality spots with ≤ 100 transcripts, ≤ 5 genes or ≥ 30% mitochondrial gene content. Genes matching regex patterns ^Rp[l|s], for ribosomal genes and ^mt- or ^Mrp[l|s] for mitochondrial genes were removed and only genes annotated as protein-coding were retained. Normalisation and scaling of the Visium data were performed using the Seurat (v4.4.0) functions NormalizeData and ScaleData. Cell type deconvolution results were downloaded from Mendeley Data (DOI: 10.17632/w7nw4km7xd.1). Spatial cross-correlation was performed between TIC-normalised MSI data, cell type scores and normalised Visium gene expressions using the spatial cross-correlation metric from MERINGUE[74] (v1.0).

**Same-section human breast cancer tumour samples.** Breast cancer tissue sections BC_1_515, BC_1_525 and BC_2_515 were previously analysed using DESI-MSI and Visium spatial transcriptomics as described in Godfrey et al.[57]. Authors from the original study are co-authors of this manuscript and contributed to data generation and interpretation and provided Visium Space Ranger outputs and MSI .imzml files for re-analysis using MAGPIE. The MSI table extracted from SCiLS Lab was then used as input to the MAGPIE pipeline using default parameters except agg_fn = 'weighted_average' and radius_to_use = 'msi' in create_barcode_matrix. Both modalities were read using semla[26] (v1.2.1). Visium data was filtered to remove low quality spots with ≤ 100 transcripts, ≤ 5 genes or ≥ 10% mitochondrial gene content. The Visium data was further filtered to remove genes matching the regex patterns ^RP[L|S], for ribosomal genes and ^MT- or ^MRP for mitochondrial genes and only genes annotated as protein-coding were retained. Normalisation and scaling of the Visium data were performed using the Seurat (v4.4.0) functions NormalizeData and ScaleData. Clustering of MSI data was performed per sample using kmeans function in R with centres = 2, then combined into cancer/non-cancer clusters across all samples. Cell type deconvolution was performed using CARD[58] (v1.1) on the Wu et al. human breast cancer atlas[59]. Radial distance analysis was performed using the RadialDistance function in semla[26] to link metabolite peak intensity, gene expression and cell type scores with

distance from cancer regions. Peak intensities and gene expressions were compared using Spearman correlation.

**Statistics & Reproducibility.** The approach was tested on 4 datasets: [1] bleomycin-treated mouse samples ($n = 11$), [2] AZX-treated rat lung sample ($n = 1$), [3] human ($n = 3$) and mouse ($n = 3$) brain tissue, and [4] human breast cancer tissue ($n = 3$). No statistical method was used to predetermine sample size as this work focused on a new methodology and demonstrating its applicability on several different tissue types.

Whole samples were excluded where tissue disruption prohibited coregistration, as detailed in Supplementary Fig. 2c. Post-coregistration Visium data was filtered using spot and gene-based thresholds as described in **Methods**. DESI MSI data was filtered using pixel-based and peak-based parameters described in **Methods**.

In the full published bleomycin study, mice were randomly assigned to treatment groups to ensure that experimental conditions were evenly distributed. For the purposes of our methodology study we selected 9 of these animals. In the full AZX-dosed rat study, rats were randomly assigned to dose groups but we selected one sample for this methodology study. The Investigators were not blinded to allocation during experiments and outcome assessment.

### Reporting summary

Further information on research design is available in the Nature Portfolio Reporting Summary linked to this article.

## Data availability

The previously generated Visium data from the bleomycin mouse model is available in Gene Expression Omnibus under accession number GSE267904; all associated metadata is available at BioStudies with accession number S-BSST1409. The newly generated MSI data from the bleomycin mouse model has been deposited in the MetaboLights database under accession code MTBLS13445. The AZX-treated rat Visium data has been deposited to the Gene Expression Omnibus database under accession code GSE312168, and the MSI data has been deposited to the MetaboLights database under accession code MTBLS13445. The mouse and human brain tissue samples from the study by Vicari et al. are deposited on Mendeley Data, (https://doi.org/10.17632/w7nw4km7xd.1). The human breast cancer data from the study by Godfrey et al. are deposited on the Harvard Dataverse with reference number 107910 (https://doi.org/10.7910/DVN/GZFCWC). All processed data for all datasets have been deposited on Zenodo under (https://doi.org/10.5281/zenodo.17789448)[76]. Source data are provided with this paper.

## Code availability

The MAGPIE source code is publicly available at (https://github.com/Core-Bioinformatics/magpie) ith a mirrored deposition to Zenodo under (https://doi.org/10.5281/zenodo.17751881[77]). All the code used to perform the downstream analyses is available on Zenodo under (https://doi.org/10.5281/zenodo.17789448)[76].

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

## Acknowledgements

This work was supported by the Swedish Foundation for Strategic Research (grant no. ID18-0094 supporting L.F.), the MRC DTP iCASE studentship programme (G117817 supporting E.C.W.), and AstraZeneca. I.M. was funded by the Wellcome Trust [203151/Z/16/Z] and the UKRI Medical Research Council [MC_PC_17230]. We thank A. Collin and E. Sand for assistance and input on tissue sample collection and assessment, A. Borde and T. Volckaert for providing the mice from the bleomycin mouse model study, B. Keith for Visium data pre-processing, M. Hühn and V. Ptasinski for initial results discussions, and R. Mauron, M. Machado, and M. Ekvall for input on multi-modal integration. For the purpose of open access, the authors applied for a CC BY public copyright license to all versions of the manuscript arising from this submission.

## Author contributions

E.C.W., L.F., J.T. and I.M. designed and implemented the pipeline and performed downstream analysis. J.D. and A.H. provided support for early versions of the co-registration pipeline. J.E.M. contributed to the design of the data mapping strategy. S.O. optimised the rodent lung sampling strategy. J.J.H., A.O., M.S., L.F., M.O.L., and M.M.M. planned the experimental work. M.O.L. sectioned the rodent lung tissues and generated the rodent Visium data. M.M.M. provided the rat lung samples. G.H. ran the MSI experiments for the rodent lung samples, and G.H., E.C.W. and A.Z. performed the MSI data pre-processing. L.S. performed the histopathological assessment of the rodent lung tissues. T.M.G. and L.S.E. provided human breast cancer data and feedback on analysis. E.C.W. and M.V. performed manual landmark selection of samples. J.E.M., M.V., and J.L. contributed to methodological discussions. Data interpretation was done by L.F., M.O.L., E.C.W. and G.H. L.F. and E.C.W. created the final figures and illustrations. E.C.W., L.F., M.O.L., and I.M. wrote the manuscript with input from other authors. J.J.H., A.O., P.L.S., M.S., J.T. and I.M. guided and supervised the project.

## Competing interests

E.C.W. is partly funded by AstraZeneca and M.O.L., G.H., S.O., J.D., A.H., M.M., L.S, J.J.H., A.O., and M.S. are employees and/or stockholders of AstraZeneca. J.T. and L.F. were AstraZeneca employees at the time of the study but are currently employed by GSK and Pixelgen Technologies AB, respectively. L.S.E. is an inventor in patents related to DESI-MS imaging technology owned by Purdue Research Foundation that were licensed to Waters Corporation and receives royalties from sales of the systems. The remaining authors declare no competing interests.
