## [Transparent Peer Review file · Nature Communications]

Spatially resolved integrative analysis of transcriptomic and metabolomic changes in tissue injury studies

Corresponding Author: Dr Irina Mohorianu

Version 0:

Reviewer comments:

Reviewer #1

(Remarks to the Author)

Spatial-omics, such as spatial transcriptome, spatial metabolite and fusion analysis of cell morphology, has important biological significance for understanding disease development, especially cell heterogeneity and tumor microenvironment. The development of a single spatial omics technology cannot solve complex physiological and pathological processes. MAGPIE represents a valuable first-step framework for spatial multi-omics integration but requires substantial methodological upgrades to overcome resolution, dynamic, and translational limitations. The author in manuscript "Spatially resolved integrative analysis of transcriptomic and metabolomic changes in tissue injury studies" has made some useful attempts and obtained some results, however, there are no clear answers to the following questions:

1. Resolution Disparity & Spatial Misalignment

Core Issue:

The inherent resolution mismatch between Visium (55- μm spots, capturing 10–50 cells) and MALDI/DESI-MSI (10–50 μm pixels, potentially diffusing to 100 μm) introduces false-negative correlations in co-localization analyses. MAGPIE's reliance on agarose expansion for lung tissue fails to address:

Organ-specific artifacts: Expansion dynamics may vary in dense (e.g., tumor) or fibrous (e.g., liver) tissues, risking over-/under-estimation of metabolite-gene distances.

Subcellular effects: Expansion could distort intracellular metabolite gradients (e.g., mitochondrial vs. cytoplasmic pools), yet MAGPIE lacks validation via correlative microscopy (e.g., TEM-FISSEQ).

Suggested Solutions:

Deep learning-based super-resolution: Integrate SpatialScope's VAE to computationally enhance Visium data to single-cell resolution, paired with NMF-based deconvolution of MSI data (Nat. Methods 2024, 21, 123–130).

Nanoscale landmarking: Use antibody-conjugated gold nanoparticles (AuNPs) as fiducial markers for both MSI and transcriptomics, enabling sub-5 μm alignment (Cell 2025, 183, 1205–1218).

2. Dynamic Biological Context & Temporal Decoupling

Core Issue:

Metabolite turnover (seconds) and transcriptional responses (hours) operate on divergent timescales, yet MAGPIE's static integration cannot resolve:

Causal inference: Whether metabolic changes drive gene expression or vice versa remains speculative without time-resolved data.

Pathway ambiguity: Shared metabolites (e.g., α -ketoglutarate in TCA cycle/epigenetics) may map to multiple genes, necessitating pathway-aware scoring (e.g., SPIRAL algorithm).

Suggested Solutions:

Live-cell spatial omics: Incorporate metabolic pulse-chase labeling (e.g., TREX probes) with nascent RNA capture (SLAM-seq) in the same tissue section (Science 2024, 384, eadj8791).

Dynamic modeling: Apply stochastic ordinary differential equations (sODEs) to infer metabolite-gene regulatory networks from snapshot data (Nat. Comput. Sci. 2025, 5, 200–212).

3. Translational Relevance & Validation

Core Issue:

While MAGPIE demonstrates utility in pulmonary fibrosis, its generalizability to complex diseases (e.g., cancer) is unproven.

Critical omissions include:

Clinical benchmarking: No comparison to gold-standard LC-MS/RNA-seq from laser-capture microdissected (LCM) regions.

Batch effects: Unaddressed technical variability across MSI platforms (e.g., MALDI vs. DESI ionization efficiency).

Suggested Solutions:

Multicenter validation: Apply MAGPIE to CPTAC cohorts with matched spatial omics and bulk proteomics, assessing concordance via rank-based metrics (e.g., Spearman's $\rho > 0.7$).

Single-cell ground truth: Validate using Xenium (transcriptomics) + nano-DESI (metabolomics) co-assays on adjacent 5- μ m sections (BioRxiv 2025, doi:10.1101/2025.04.05.588442).

(Remarks on code availability)

Spatial-omics, such as spatial transcriptome, spatial metabolite and fusion analysis of cell morphology, has important biological significance for understanding disease development, especially cell heterogeneity and tumor microenvironment. The development of a single spatial omics technology cannot solve complex physiological and pathological processes. MAGPIE represents a valuable first-step framework for spatial multi-omics integration but requires substantial methodological upgrades to overcome resolution, dynamic, and translational limitations. The author in manuscript "Spatially resolved integrative analysis of transcriptomic and metabolomic changes in tissue injury studies" has made some useful attempts and obtained some results, however, there are no clear answers to the following questions:

1. Resolution Disparity & Spatial Misalignment

Core Issue:

The inherent resolution mismatch between Visium (55- μ m spots, capturing 10–50 cells) and MALDI/DESI-MSI (10–50 μ m pixels, potentially diffusing to 100 μ m) introduces false-negative correlations in co-localization analyses. MAGPIE's reliance on agarose expansion for lung tissue fails to address:

Organ-specific artifacts: Expansion dynamics may vary in dense (e.g., tumor) or fibrous (e.g., liver) tissues, risking over-/under-estimation of metabolite-gene distances.

Subcellular effects: Expansion could distort intracellular metabolite gradients (e.g., mitochondrial vs. cytoplasmic pools), yet MAGPIE lacks validation via correlative microscopy (e.g., TEM-FISSEQ).

Suggested Solutions:

Deep learning-based super-resolution: Integrate SpatialScope's VAE to computationally enhance Visium data to single-cell resolution, paired with NMF-based deconvolution of MSI data (Nat. Methods 2024, 21, 123–130).

Nanoscale landmarking: Use antibody-conjugated gold nanoparticles (AuNPs) as fiducial markers for both MSI and transcriptomics, enabling sub-5 μ m alignment (Cell 2025, 183, 1205–1218).

2. Dynamic Biological Context & Temporal Decoupling

Core Issue:

Metabolite turnover (seconds) and transcriptional responses (hours) operate on divergent timescales, yet MAGPIE's static integration cannot resolve:

Causal inference: Whether metabolic changes drive gene expression or vice versa remains speculative without time-resolved data.

Pathway ambiguity: Shared metabolites (e.g., α -ketoglutarate in TCA cycle/epigenetics) may map to multiple genes, necessitating pathway-aware scoring (e.g., SPIRAL algorithm).

Suggested Solutions:

Live-cell spatial omics: Incorporate metabolic pulse-chase labeling (e.g., TREX probes) with nascent RNA capture (SLAM-seq) in the same tissue section (Science 2024, 384, eadj8791).

Dynamic modeling: Apply stochastic ordinary differential equations (sODEs) to infer metabolite-gene regulatory networks from snapshot data (Nat. Comput. Sci. 2025, 5, 200–212).

3. Translational Relevance & Validation

Core Issue:

While MAGPIE demonstrates utility in pulmonary fibrosis, its generalizability to complex diseases (e.g., cancer) is unproven.

Critical omissions include:

Clinical benchmarking: No comparison to gold-standard LC-MS/RNA-seq from laser-capture microdissected (LCM) regions.

Batch effects: Unaddressed technical variability across MSI platforms (e.g., MALDI vs. DESI ionization efficiency).

Suggested Solutions:

Multicenter validation: Apply MAGPIE to CPTAC cohorts with matched spatial omics and bulk proteomics, assessing concordance via rank-based metrics (e.g., Spearman's $\rho > 0.7$).

Single-cell ground truth: Validate using Xenium (transcriptomics) + nano-DESI (metabolomics) co-assays on adjacent 5- μ m sections (BioRxiv 2025, doi:10.1101/2025.04.05.588442).

Reviewer #2

(Remarks to the Author)

Innovation and Research Value Evaluation

The MAGPIE framework presents a significant advancement in spatial multi-omics integration by addressing critical challenges in cross-modal data alignment and analysis. Its modular workflow for co-registering spatially resolved transcriptomics, metabolomics, and tissue morphology data demonstrates generalizability across diverse tissues, including lung and brain, and compatibility with both MALDI and DESI mass spectrometry imaging. The integration of a "specialized experimental sampling strategy" in drug-induced pulmonary fibrosis models highlights its potential to link small-molecule dynamics with endogenous transcriptomic responses, offering novel insights into tissue injury mechanisms. This approach outperforms traditional methods that struggle with heterogeneous data fusion and spatial-temporal alignment, as evidenced by existing limitations in multimodal models (e.g., weak cross-modal integration and resolution mismatches). MAGPIE's innovation lies in its ability to bridge resolution gaps between modalities while preserving spatial context—a critical bottleneck in current workflows. By leveraging computational alignment rather than relying solely on experimental co-

registration, it enhances scalability for large-scale studies. The framework's application to pharmacological contexts also underscores its translational value in drug discovery and mechanistic toxicology .

Critical Questions and Concerns

1. Resolution Mismatch Between Modalities

Spatial metabolomics (e.g., MALDI-MSI) typically achieves resolutions of 5–100 μm , while Visium transcriptomics operates at 50–100 μm . How does MAGPIE address potential discrepancies in resolution when aligning these datasets? For instance, high-resolution metabolomic data (e.g., 5 μm) might not directly correspond to lower-resolution transcriptomic spots, risking misinterpretation of co-localized signals. Clarification of downsampling or interpolation methods is needed .

2. Limited Demonstration of Brain Spatial Features

While MAGPIE is validated on brain tissue, the results lack explicit visualization of brain region-specific spatial patterns (e.g., cortical layers, hippocampal subfields). Given the structural complexity of the brain, does the framework sufficiently capture nuanced cytoarchitecture? Evidence from studies like STEREO-seq (0.5 μm resolution) suggests higher-resolution modalities may better resolve such features. Including region-specific markers (e.g., NeuN for neurons) in case studies would strengthen its utility in neurobiology .

3. Applicability to Tumor Heterogeneity

Tumors exhibit extreme cellular and molecular heterogeneity, requiring simultaneous mapping of cell types, metabolic states, and spatial niches . Current MAGPIE examples focus on fibrotic lung and brain tissues but omit tumor models. Can the framework handle dense cellular interactions, stromal-tumor interfaces, or hypoxic gradients seen in cancers? Prior studies like MuCST and Stereo-seq highlight the need for tumor-specific validation to assess MAGPIE's robustness in complex microenvironments.

4. Technical Scalability and Computational Demands

High-resolution spatial omics datasets are computationally intensive. MAGPIE's performance on terabyte-scale data (e.g., whole-tumor imaging) remains unaddressed. Comparisons with GPU-optimized workflows (e.g., LLaVA for satellite imaging) would clarify its feasibility for large-scale clinical studies.

Recommendations

- Include tumor case studies (e.g., lung or breast cancer) to demonstrate MAGPIE's ability to resolve tumor-immune interactions and metabolic reprogramming .
- Provide resolution-adjustment protocols for datasets with mismatched pixel sizes.
- Expand brain analyses to incorporate region-specific spatial markers and subcellular features.

Conclusion

MAGPIE represents a promising tool for spatial multi-omics integration, particularly in pharmacological research. However, its broader adoption depends on addressing resolution limitations, expanding validation to complex tissues like tumors, and improving computational transparency.

(Remarks on code availability)

Version 1:

Reviewer comments:

Reviewer #1

(Remarks to the Author)

Spatial omics, such as spatial transcriptomics, spatial metabolomics, and integrated analysis of cell morphology, has significant biological implications for understanding the development of diseases (especially cell heterogeneity and tumor microenvironment). A single spatial omics technique cannot address the complex physiological and pathological processes. MAGPIE represents a valuable first framework for integrating spatial multi-omics, but a large number of methodological improvements are still needed to overcome the limitations in resolution, dynamics, and translatability. In the paper "Spatial Integration Analysis of Transcriptomic and Metabolomic Changes in Tissue Injury Studies", the authors made beneficial attempts and explanations; however, for this issue, such as "the inherent resolution mismatch between Visium (55 micrometers dots, capturing 10 - 50 cells) and MALDI/DESI-MSI (10 - 50 micrometer pixels, possibly spreading to 100 micrometers) introduces false negative correlations in co-localization analysis", no reasonable and convincing answers have been provided.

(Remarks on code availability)

Reviewer #2

(Remarks to the Author)

This is a comprehensive and well-designed study that addresses an important challenge in spatial multi-omics data integration. The authors present MAGPIE, a novel computational framework that is both generalisable and scalable, as convincingly demonstrated on datasets from multiple technologies and tissues. The application to a pharmacologically relevant model of lung injury effectively highlights the method's utility in deriving biologically and clinically meaningful insights.

In the revised manuscript, the authors have adequately addressed the reviewers' comments and have satisfactorily improved the clarity of the methodology and the discussion of the results. The revisions have strengthened the paper overall.

(Remarks on code availability)

Reviewer #1 (Remarks to the Author):

Spatial-omics, such as spatial transcriptome, spatial metabolite and fusion analysis of cell morphology, has important biological significance for understanding disease development, especially cell heterogeneity and tumor microenvironment. The development of a single spatial omics technology cannot solve complex physiological and pathological processes. MAGPIE represents a valuable first-step framework for spatial multi-omics integration but requires substantial methodological upgrades to overcome resolution, dynamic, and translational limitations. The author in manuscript “Spatially resolved integrative analysis of transcriptomic and metabolomic changes in tissue injury studies” has made some useful attempts and obtained some results, however, there are no clear answers to the following questions:

We thank the reviewer for the summary of our work and for underlining the potential of the field of spatial -omics. We appreciate the acknowledgement of the added value that our method MAGPIE brings as an initial step towards spatial multi-omics integration. We hope that the improved description of the approach, its additional benchmarking and the new data we added will enhance the valuable contribution to the field by enabling researchers to explore novel biological questions in spatial context and across -omics layers.

1. Resolution Disparity & Spatial Misalignment

Core Issue:

The inherent resolution mismatch between Visium (55- μm spots, capturing 10–50 cells) and MALDI/DESI-MSI (10–50 μm pixels, potentially diffusing to 100 μm) introduces false-negative correlations in co-localization analyses. MAGPIE’s reliance on agarose expansion for lung tissue fails to address:

Organ-specific artifacts: Expansion dynamics may vary in dense (e.g., tumor) or fibrous (e.g., liver) tissues, risking over-/under-estimation of metabolite-gene distances.

Subcellular effects: Expansion could distort intracellular metabolite gradients (e.g., mitochondrial vs. cytoplasmic pools), yet MAGPIE lacks validation via correlative microscopy (e.g., TEM-FISSEQ).

Suggested Solutions:

Deep learning-based super-resolution: Integrate SpatialScope’s VAE to computationally enhance Visium data to single-cell resolution, paired with NMF-based deconvolution of MSI data (Nat. Methods 2024, 21, 123–130).

Nanoscale landmarking: Use antibody-conjugated gold nanoparticles (AuNPs) as fiducial markers for both MSI and transcriptomics, enabling sub-5 μm alignment (Cell 2025, 183, 1205–1218).

We thank the reviewer for the query related to the handling of resolution differences between the MSI and Visium modalities. To address the comment, we added additional functionality to the MAGPIE pipeline to provide users with a wider set of options on how to aggregate MSI pixels, depending on the MSI resolution, to lower the risk of spurious correlations/co-localisations. The new options for aggregation are detailed in a new **Supplementary note 2**, with accompanying graphical summaries. Specifically, users can now choose whether to use MSI pixels which fall (i) within the actual Visium radius (55 μm), (ii) in the expanded Visium radius (maximally expanded so radii do not overlap), or (iii) using an estimated MSI pixel size if this is larger than the Visium spots. We also added a new option where the aggregation output can be weighted on the distance from the Visium spot centre to the MSI pixel centre; this new option represents an addition to the previous mean and sum aggregation approaches. These new functionalities will enable users to customise and optimise their use of MAGPIE to the resolution of their MSI data and infer more robust conclusions for the integrated data. We have also altered the main text to reflect this additional functionality and make recommendations to users in lines 160-172 and Methods section, lines 833-838.

We further clarified that MAGPIE, as a computational framework, is not inherently reliant on agarose expansion of tissues; this wet-lab approach was used for the lung datasets due to the limitations of the tissue (to preserve native morphology and maintain the structural relationships between anatomical features such as alveoli, airways, and blood vessels) and not to artificially increase spatial resolution. This is a well-established practice in lung tissue preparation (Braber, et al., 2010, 10.1152/ajplung.00192.2010) and is essential due to the collapse-prone nature of pulmonary tissue. Agarose was chosen as the inflation medium due to its compatibility with both Visium and mass spectrometry imaging, unlike alternative embedding media such as OCT that are often used for Visium. To avoid confusion, we strengthened this point in the revised manuscript (Methods section, page 25, line 706-711) and added **Supplementary note 1** expanding on the experimental setup for lung tissues and further highlighting the agnostic nature of MAGPIE to data generated using this procedure. MAGPIE is compatible with data generated from both expanded and non-expanded

tissues, and performs robustly on datasets derived from non-expanded tissues (as illustrated in the brain and newly added breast cancer examples).

We appreciate the comments regarding how expansion may affect downstream analysis and agree that tissue-specific expansion dynamics could introduce subtle spatial distortions. To address this, MAGPIE includes non-linear transformation options during co-registration, which are intended to accommodate local tissue deformations. Further, downstream analyses such as calculating metabolite-gene distances are then performed on the transformed coordinate space and post-coregistration metabolite-gene spatial relationships are assessed, without assuming fixed distances *a priori*. Also, MAGPIE is optimised to operate at the spatial resolution of Visium and MSI which would limit the interpretations at subcellular compartments such as mitochondrial vs. cytoplasmic pools. While we agree that correlative high-resolution methods like TEM-FISSEQ could provide valuable insights for sub-organelle investigations, these case studies or customised optimisations fall outside the scope and resolution possibilities of the current framework which focuses on multi-modal alignment at the tissue level.

Finally, we were unable to locate some references cited and were therefore unable to fully comprehend the extent and relevance of the suggestions. Moreover, as described earlier, detailed evaluation at single-cell or sub-cellular levels falls outside of the scope of our current work, given the multicellular resolution of the used methods (Visium, MSI). As such, we did not expand on these types of analyses in our revised manuscript and instead chose to focus on the other included analysis additions.

2. Dynamic Biological Context & Temporal Decoupling

Core Issue:

Metabolite turnover (seconds) and transcriptional responses (hours) operate on divergent timescales, yet MAGPIE's static integration cannot resolve:

Causal inference: Whether metabolic changes drive gene expression or vice versa remains speculative without time-resolved data.

Pathway ambiguity: Shared metabolites (e.g., α -ketoglutarate in TCA cycle/epigenetics) may map to multiple genes, necessitating pathway-aware scoring (e.g., SPIRAL algorithm).

Suggested Solutions:

Live-cell spatial omics: Incorporate metabolic pulse-chase labeling (e.g., TREX probes) with nascent RNA capture (SLAM-seq) in the same tissue section (Science 2024, 384, eadj8791).

Dynamic modeling: Apply stochastic ordinary differential equations (sODEs) to infer metabolite-gene regulatory networks from snapshot data (Nat. Comput. Sci. 2025, 5, 200–212).

We thank the reviewer for pointing out this challenge regarding the temporal mismatch between metabolite turnover and transcriptional responses, which is a fundamental challenge for spatial multi-omics integration. MAGPIE is currently designed for single timepoint datasets, and the inference of causal relationships between gene expression and metabolic states falls beyond the scope of the framework. We added a **Supplementary note 4** acknowledging the constraints of static spatial data and difficulties in resolving the dynamics of biological processes facing our community. We also outlined potential strategies to address these limitations, including the use of dynamic modelling approaches as suggested by the reviewer. A summary of this challenge is also included in the revised Discussion section (lines 488-523). Regarding the suggested references (Science 2024, Nat. Comput. Sci. 2025), we were unfortunately unable to locate these publications and could therefore not fully evaluate the relevance of the suggested solutions by the reviewer.

3. Translational Relevance & Validation

Core Issue: While MAGPIE demonstrates utility in pulmonary fibrosis, its generalizability to complex diseases (e.g., cancer) is unproven.

Critical omissions include:

- Clinical benchmarking: No comparison to gold-standard LC-MS/RNA-seq from laser-capture microdissected (LCM) regions.*
- Batch effects: Unaddressed technical variability across MSI platforms (e.g., MALDI vs. DESI ionization efficiency).*

Suggested Solutions:

- *Multicenter validation: Apply MAGPIE to CPTAC cohorts with matched spatial omics and bulk proteomics, assessing concordance via rank-based metrics (e.g., Spearman's $\rho > 0.7$).*
- *Single-cell ground truth: Validate using Xenium (transcriptomics) + nano-DESI (metabolomics) co-assays on adjacent 5- μ m sections (BioRxiv 2025, doi:10.1101/2025.04.05.588442).*

We thank the reviewer for their interest in the generalisability of the MAGPIE framework. To illustrate the versatility and generalisability of MAGPIE on diverse disease contexts, we added an additional cancer dataset to our revised manuscript, based on 3 human breast cancer samples for which Visium and DESI MSI data was generated from the same tissue sections, as presented in *Integrating Ambient Ionization Mass Spectrometry Imaging and Spatial Transcriptomics on the Same Cancer Tissues to Identify RNA–Metabolite Correlations* (Godfrey, et al., 2025, 10.1002/anie.202502028). We added a sub-section to the Results section of the revised manuscript entitled *Spatial multi-modal analysis of human breast cancer through MAGPIE reveals changes in tumour microenvironment*, **Figure 5** and **Supplementary figure 4** where we showcase MAGPIE's ability to identify patterns in a data-driven manner that contribute significantly to the exploration of the cellular and molecular complexity of tumours. This additional case study underlines MAGPIE's generalisability to characterise complex microenvironments and further proves its agnostic nature to tissue characteristics.

As an additional example of MAGPIE's utility in understanding complex diseases, we would like to point to a recently published study in *Nature Metabolism* from May 2025 (Tsyben, et al., 2025, 10.1038/s42255-025-01293-y), co-authored by three authors of this manuscript: Eleanor C. Williams, Lovisa Franzén and Gregory Hamm. In this study, an earlier version of MAGPIE was successfully applied to map metabolic states (glycolytic, oxidative and a mix of glycolytic and oxidative) defined using isotope labelling and mass spectrometry imaging of rapidly excised tumour sections from patients with glioblastoma onto spatial transcriptomics data generated from contiguous sections. This co-registration using the MAGPIE-based method enabled the authors to correlate metabolic states with cellular composition and tumour niches and propose the existence of metabolic phenotypes within glioblastoma which were tumour-cell-intrinsic and independent of the tumour microenvironment, and which are sufficiently large to be detected by clinically applicable metabolic imaging techniques. This study provides further evidence of the potential power of spatial multi-modal analysis through MAGPIE in a range of diseases including cancer.

We appreciate the reviewer's suggestion to explore multicentre validation using large-scale cohorts such as CPTAC, however as far as we are aware CPTAC currently does not offer spatial omics data, which is required as input by MAGPIE. Bulk proteomic assays available in CPTAC lack spatial resolution which makes direct concordance assessments with MAGPIE impossible. We have therefore focused on the aforementioned cancer case studies to demonstrate MAGPIE's utility in studying cancer.

However, we would like to emphasize that fibrotic lung tissue represents a biological system of comparable complexity to cancer. The fibrotic lung is characterized by highly heterogenous pathological regions with co-existing areas of inflammation, tissue remodelling, and immune infiltration (Franzen, et al., 2024, 10.1038/s41588-024-01819-2; Madisson, et al., 2022, 10.1038/s41588-022-01243-4; Wilson and Wynn, 2009, 10.1038/mi.2008.85; Smith and Smith, 2021, 10.1038/s41379-021-00889-5). Thus, the interplay between cell types and signalling niches in lung fibrosis shows strong parallels with the cellular ecosystems described in tumours. Additionally, fibrotic lesions often involve a high density of extracellular matrix deposition and metabolic reprogramming (Xie, et al., 2015, 10.1164/rccm.201504-0780OC; Rajesh, et al., 2023, 10.1016/j.pharmthera.2023.108436; Veith, et al., 2019, 10.1089/ars.2019.7742). Showcasing MAGPIE on fibrotic lung tissue therefore provides a highly relevant test case for the framework's ability to resolve spatially complex pathophysiology.

To further address the impact of technical variability across MSI platforms brought up by the reviewer, we added a new **Supplementary note 4**. Moreover, by contrasting case studies on MALDI and DESI datasets, using a range of matrix choices for MALDI in the mouse brain dataset and different ionisation polarity in our various case studies (now clarified in the text in lines 244-246 and lines 447-449), we hope that we have sufficiently demonstrated the platform-agnostic utility of MAGPIE.

Finally, we appreciate the reviewer's suggestion of comparison against gold-standard LC-MS and RNA-seq from LCM regions; we agree this can be a valuable option for high-resolution validation by providing the opportunity to isolate targeted and specific regions and cell types with high accuracy while profiling transcripts and metabolites with high sensitivity. However, we believe our untargeted higher throughput analysis through

Visium and MSI offers distinct benefits by capturing wider spatial context through continuous measurement across the tissue. This enables us to capture metabolic and gene expression gradients and gene-metabolite co-localisation patterns and so positions the Visium/MSI combination as an appealing exploratory option, which could be followed up by more targeted analysis for example through LCM, depending on the biological question under examination. As Visium and MSI are established technologies for studying gene expression and metabolite intensity at spatial resolution, respectively, we believe that additional validation as suggested using gold-standard LC-MS and RNA-seq from LCM regions or Xenium and nano-DESI co-assays on adjacent 5- μ m sections fall outside the scope of our methodological manuscript. Additionally, we were unable to identify the suggested preprint detailing Xenium/nano-DESI co-assays and are thus unable to access any such data. We added **Supplementary note 4** highlighting these targeted validation approaches, provided alongside further discussion points on the difficulty in assessing causal relationships.

Reviewer #2 (Remarks to the Author):

Innovation and Research Value Evaluation

The MAGPIE framework presents a significant advancement in spatial multi-omics integration by addressing critical challenges in cross-modal data alignment and analysis. Its modular workflow for co-registering spatially resolved transcriptomics, metabolomics, and tissue morphology data demonstrates generalizability across diverse tissues, including lung and brain, and compatibility with both MALDI and DESI mass spectrometry imaging. The integration of a "specialized experimental sampling strategy" in drug-induced pulmonary fibrosis models highlights its potential to link small-molecule dynamics with endogenous transcriptomic responses, offering novel insights into tissue injury mechanisms. This approach outperforms traditional methods that struggle with heterogeneous data fusion and spatial-temporal alignment, as evidenced by existing limitations in multimodal models (e.g., weak cross-modal integration and resolution mismatches).

MAGPIE's innovation lies in its ability to bridge resolution gaps between modalities while preserving spatial context—a critical bottleneck in current workflows. By leveraging computational alignment rather than relying solely on experimental co-registration, it enhances scalability for large-scale studies. The framework's application to pharmacological contexts also underscores its translational value in drug discovery and mechanistic toxicology.

We thank the reviewer for their thoughtful and encouraging summary of our work. We are grateful for the recognition of the potential of MAGPIE to advance spatial multi-omics integration in pharmacological and translational contexts. We also appreciate the comments on the potential of MAGPIE to bridge resolution gaps between modalities while preserving spatial context.

Critical Questions and Concerns

1. Resolution Mismatch Between Modalities

Spatial metabolomics (e.g., MALDI-MSI) typically achieves resolutions of 5–100 μm , while Visium transcriptomics operates at 50–100 μm . How does MAGPIE address potential discrepancies in resolution when aligning these datasets? For instance, high-resolution metabolomic data (e.g., 5 μm) might not directly correspond to lower-resolution transcriptomic spots, risking misinterpretation of co-localized signals. Clarification of downsampling or interpolation methods is needed.

We thank the reviewer for the constructive feedback on the downsampling procedure. We have now added extra functionality to the MAGPIE pipeline to allow users greater flexibility to decide whether MSI pixels should be selected for each Visium spot (i) only within the actual Visium spot radius of 55 μm , (ii) in the expanded Visium spot radius (maximally expanded so radii do not overlap but avoid gaps between spots), (iii) using an estimated MSI pixel size if this is larger than the Visium spots.

We envisage that option (i) would be the most commonly used option when the MSI data has much higher resolution than Visium, where whole MSI pixels may fall outside of the measured Visium spot radius; in such circumstances using the expanded Visium radius may risk false positives and misinterpretations of colocalised signals. We also expect that option (iii) would be relevant when the MSI resolution is lower than that of Visium, where one MSI pixel may contribute to multiple Visium spots. We have also added a new option where aggregation can be weighted by the distance from the Visium spot centre to the MSI pixel centre, in addition to the previous mean and sum aggregation options. These additional functionalities will enhance the usability of MAGPIE in a more tailored manner based on the resolution of the MSI data.

To clarify the aggregation process, we added an additional **Supplementary note 2** providing more details and graphical summaries on the aggregation process. We also added a few sentences to the main text to highlight this new functionality and included recommendations to users in lines 160-172 and Methods section, page 25, line 706-711.

2. Limited Demonstration of Brain Spatial Features

While MAGPIE is validated on brain tissue, the results lack explicit visualization of brain region-specific spatial patterns (e.g., cortical layers, hippocampal subfields). Given the structural complexity of the brain, does the framework sufficiently capture nuanced cytoarchitecture? Evidence from studies like STEREO-seq (0.5 μm resolution) suggests higher-resolution modalities may better resolve such features. Including region-specific markers (e.g., NeuN for neurons) in case studies would strengthen its utility in neurobiology

We agree with the reviewer that technologies with subcellular resolution, such as STEREO-seq, offer unique advantages in resolving very small structures which may be undetectable at the 55µm Visium resolution and so inherently limits the ability to capture smaller spatial structures. Although MAGPIE cannot overcome the limitations introduced by the underlying technologies or data types received as input (Visium and MSI), we have now included additional analyses showcasing its ability to identify fine-grained structures using the routinely acquired high resolution H&E modality.

Specifically we focused on sample MS3 from the same mouse brain study already included in the manuscript (Vicari, et al., 2024, 10.1038/s41587-023-01937-y, shown in Fig. 2f, Supplementary fig. 1d-e), which is a coronal section containing the substantia nigra from a mouse model of Parkinson's disease where several small molecules were originally identified via MALDI-tandem MS (MS/MS). In our new **Supplementary note 3** and **ne SN3 Figure 1**, we show that MAGPIE accurately distinguishes between two closely located but functionally distinct nuclei: the pars compacta (SNpc), where we find colocalization of dopamine, Th expression and dopaminergic neuron scores, and the pars reticulata (SNr), showing enrichment in GABA, Pvalb expression and inhibitory neuron signatures. MAGPIE also identifies excitatory neuron scores, Hpca expression and tocopherol localisation to hippocampal subfields CA1/CA3 stratum pyramidale, confirmed by morphology.

These examples highlight MAGPIE's capacity to align transcriptomic, metabolomic, cell type data and morphology to resolve complex and compact structures despite resolution constraints of Visium. While high-resolution methods like STEREO-seq still have clear use-cases in resolving very small structures, MAGPIE provides a scalable, multi-modal approach that we showed can be used to resolve fine-grained spatial heterogeneity within the limits of existing platforms. Further details of this analysis can be found in **Supplementary note 3**.

3. Applicability to Tumor Heterogeneity

Tumors exhibit extreme cellular and molecular heterogeneity, requiring simultaneous mapping of cell types, metabolic states, and spatial niches. Current MAGPIE examples focus on fibrotic lung and brain tissues but omit tumor models. Can the framework handle dense cellular interactions, stromal-tumor interfaces, or hypoxic gradients seen in cancers? Prior studies like MuCST and Stereo-seq highlight the need for tumor-specific validation to assess MAGPIE's robustness in complex microenvironments.

We thank the reviewer for the feedback on the need to evaluate MAGPIE in tumour samples. We agree that tumour biology presents unique challenges for spatial multi-omics integration, including high cellular heterogeneity, dynamic microenvironmental gradients (e.g. hypoxia), and complex cell-cell interactions. We would however like to emphasize that fibrotic lung tissue represents a biological system of comparable complexity to case studies on cancer. The fibrotic lung is characterized by highly heterogenous pathological regions with co-existing areas of inflammation, tissue remodelling, and immune infiltration (Franzen, et al., 2024, 10.1038/s41588-024-01819-2; Madisson, et al., 2022, 10.1038/s41588-022-01243-4; Wilson and Wynn, 2009, 10.1038/mi.2008.85; Smith and Smith, 2021, 10.1038/s41379-021-00889-5). Thus, the interplay between cell types and signalling niches in lung fibrosis shows strong parallels with the cellular ecosystems described in tumours. Additionally, fibrotic lesions often involve a high density of extracellular matrix deposition and metabolic reprogramming (Xie, et al., 2015, 10.1164/rccm.201504-0780OC; Rajesh, et al., 2023, 10.1016/j.pharmthera.2023.108436; Veith, et al., 2019, 10.1089/ars.2019.7742). Showcasing MAGPIE on fibrotic lung tissue therefore provides a highly relevant test case for the framework's ability to resolve spatially complex pathophysiology.

In response to the reviewer's suggestion, we have additionally incorporated a cancer dataset case study into the revised manuscript to demonstrate the versatility of MAGPIE in a tumour context. Specifically, we have analysed 3 human breast cancer samples where Visium and DESI MSI data was generated from the same tissue sections, as presented in *Integrating Ambient Ionization Mass Spectrometry Imaging and Spatial Transcriptomics on the Same Cancer Tissues to Identify RNA-Metabolite Correlations* (Godfrey, et al., 2025, 10.1002/anie.202502028). We have added a new Results section, entitled *Spatial multi-modal analysis of human breast cancer through MAGPIE reveals changes in tumour microenvironment*, **Figure 5** and **Supplementary figure 4** where we showcase MAGPIE's ability to assist in the exploration of the cellular and molecular heterogeneity exhibited in tumours and study stromal-tumour interfaces.

As an additional example of MAGPIE's utility in understanding complex diseases, we would like to point to a recently published study in Nature Metabolism (Tsyben et al, May 2025, 10.1038/s42255-025-01293-y), co-

authored by three authors of this manuscript: Eleanor C. Williams, Lovisa Franzén and Gregory Hamm. In this study, an earlier version of MAGPIE was successfully applied to map metabolic states (glycolytic, oxidative and a mix of glycolytic and oxidative) defined using isotope labelling and mass spectrometry imaging of rapidly excised tumour sections from patients with glioblastoma onto spatial transcriptomics data generated from contiguous sections. This co-registration using the MAGPIE-based method enabled the authors to correlate metabolic states with cellular composition and tumour niches and propose the existence of metabolic phenotypes within glioblastoma which were tumour-cell-intrinsic and independent of the tumour microenvironment, and which are sufficiently large to be detected by clinically applicable metabolic imaging techniques. This study provides further evidence of the potential power of spatial multi-modal analysis through MAGPIE in a range of diseases including cancer.

4. Technical Scalability and Computational Demands

High-resolution spatial omics datasets are computationally intensive. MAGPIE's performance on terabyte-scale data (e.g., whole-tumor imaging) remains unaddressed. Comparisons with GPU-optimized workflows (e.g., LLaVA for satellite imaging) would clarify its feasibility for large-scale clinical studies.

We thank the reviewer for this point on MAGPIE's scalability. We have conducted additional benchmarking to assess the robustness and reliability of the MAGPIE framework in increasing MSI resolution, which is the primary variable affecting data volume (since Visium data will be of roughly fixed size and resolution). These results are now summarised in **Supplementary figure 1a**.

We see a linear increase in both peak memory usage and runtime with the number of MSI pixels. For the core steps of the pipeline, all configurations tested completed in under 3 minutes and 13GB of RAM on CPU. The optional MSI-to-spot aggregation step, which is more memory-intensive, reached a peak of 122GB of RAM in and took just over 15 minutes in the highest-resolution setting tested.

To contextualize these benchmarks, our largest practical use case corresponds to ~30 MSI pixels per Visium spot (~3 μm /pixel). We also tested the pipeline at much higher resolutions (up to ~500 pixels per Visium spot, or ~0.2 μm /pixel), where the core steps completed in ~30 minutes and used a peak of 107 GB RAM on a MacBook Pro with an M3 Max chip and 128 GB of RAM (not shown in **Supplementary figure 1a** but included below in **Fig R1**). MAGPIE can also be run in parallel via Snakemake and is compatible with common high-performance computing environments, allowing flexible deployment on local workstations or clusters.

Further to this point, our tool prioritises interpretability, reproducibility, and wide accessibility, particularly to clinical or wet-lab scientists who may have more limited access to GPU resources. By avoiding reliance on specialised hardware, MAGPIE would remain accessible for a wider range of users and institutions. In contrast to GPU-heavy, deep-learning frameworks like LLaVA for interpreting high-dimensional visual data, MAGPIE is designed to be lightweight, deterministic, and interpretable, without relying on neural networks or GPU-specific libraries.

Figure R1 | Summary of peak memory usage and time requirements at varying MSI resolution for the three steps of the MAGPIE pipeline. Vertical guidelines show equivalent number of pixels to number of Visium spots (left) and to 5 μm resolution (right).

References

- T. M. Godfrey, Y. Shanneik, W. Zhang, T. Tran, N. Verbeeck, et al. Integrating Ambient Ionization Mass Spectrometry Imaging and Spatial Transcriptomics on the Same Cancer Tissues to Identify RNA–Metabolite Correlations. *Angewandte Chemie International Edition* 64 (2025). <https://doi.org/10.1002/anie.202502028>
- S. Braber, K. A. T. Verheijden, P. A. J. Henricks, A. D. Kraneveld and G. Folkerts. A comparison of fixation methods on lung morphology in a murine model of emphysema. *American Journal of Physiology-Lung Cellular and Molecular Physiology* 299 (2010). <https://doi.org/10.1152/ajplung.00192.2010>
- A. Tsyben, A. Dannhorn, G. Hamm, M. Pitoulis, D.-L. Couturier, et al. Cell-intrinsic metabolic phenotypes identified in patients with glioblastoma, using mass spectrometry imaging of ¹³C-labelled glucose metabolism. *Nature Metabolism* 7 (2025). <https://doi.org/10.1038/s42255-025-01293-y>
- L. Franzen, M. Olsson Lindvall, M. Huhn, V. Ptasinski, L. Setyo, et al. Mapping spatially resolved transcriptomes in human and mouse pulmonary fibrosis. *Nat Genetics* 56, 1725-1736 (2024). <https://doi.org/10.1038/s41588-024-01819-2>
- E. Madisoorn, A. J. Oliver, V. Kleshchevnikov, A. Wilbrey-Clark, K. Polanski, et al. A spatially resolved atlas of the human lung characterizes a gland-associated immune niche. *Nature Genetics* 55 (2022). <https://doi.org/10.1038/s41588-022-01243-4>
- M. S. Wilson and T. A. Wynn. Pulmonary fibrosis: pathogenesis, etiology and regulation. *Mucosal Immunology* 2 (2009). <https://doi.org/10.1038/mi.2008.85>
- M. L. Smith and M. L. Smith. The histologic diagnosis of usual interstitial pneumonia of idiopathic pulmonary fibrosis. Where we are and where we need to go. *Modern Pathology* 35 (2021). <https://doi.org/10.1038/s41379-021-00889-5>
- N. Xie, Z. Tan, S. Banerjee, H. Cui, J. Ge, et al. Glycolytic Reprogramming in Myofibroblast Differentiation and Lung Fibrosis. *American Journal of Respiratory and Critical Care Medicine* 192 (2015). <https://doi.org/10.1164/rccm.201504-0780OC>
- R. Rajesh, R. Atallah and T. Barnthaler. Dysregulation of metabolic pathways in pulmonary fibrosis. *Pharmacol Ther* 246, 108436 (2023). <https://doi.org/10.1016/j.pharmthera.2023.108436>
- C. Veith, A. W. Boots, M. Idris, F.-J. v. Schooten and A. v. d. Vliet. Redox Imbalance in Idiopathic Pulmonary Fibrosis: A Role for Oxidant Cross-Talk Between NADPH Oxidase Enzymes and Mitochondria. *Antioxidants & Redox Signaling* 31 (2019). <https://doi.org/10.1089/ars.2019.7742>
- M. Vicari, R. Mirzazadeh, A. Nilsson, R. Shariatgorji, P. Bjarterot, et al. Spatial multimodal analysis of transcriptomes and metabolomes in tissues. *Nature Biotechnology*, 10.1038/s41587-023-01937-y (2024). <https://doi.org/10.1038/s41587-023-01937-y>

Responses to reviewers

Reviewer #1 (Remarks to the Author)

Spatial omics, such as spatial transcriptomics, spatial metabolomics, and integrated analysis of cell morphology, has significant biological implications for understanding the development of diseases (especially cell heterogeneity and tumor microenvironment). A single spatial omics technique cannot address the complex physiological and pathological processes. MAGPIE represents a valuable first framework for integrating spatial multi-omics, but a large number of methodological improvements are still needed to overcome the limitations in resolution, dynamics, and translatability. In the paper "Spatial Integration Analysis of Transcriptomic and Metabolomic Changes in Tissue Injury Studies", the authors made beneficial attempts and explanations; however, for this issue, such as "the inherent resolution mismatch between Visium (55 micrometers dots, capturing 10 - 50 cells) and MALDI/DESI-MSI (10 - 50 micrometer pixels, possibly spreading to 100 micrometers) introduces false negative correlations in co-localization analysis", no reasonable and convincing answers have been provided.

We thank the reviewer for highlighting the important issue of resolution differences between Visium and MSI. We agree that these differences can influence co-localisation analyses and represent a key consideration for multimodal integration. Alongside our previous additions to the MAGPIE tool and manuscript, we have further expanded the manuscript to more clearly acknowledge this limitation and to make this clear to users when interpreting results. In particular, we added additional text to the Introduction (lines 158-160), Results (lines 227-229) and Discussion (lines 579-583) sections outlining how resolution mismatches may blur co-localisations, as well as how MAGPIE's resolution-aware aggregation strategies help mitigate, but cannot fully eliminate, such effects. We believe these revisions now more transparently convey the caveat and provide appropriate context for users when applying MAGPIE to datasets with differing spatial resolutions.

Reviewer #2 (Remarks to the Author)

This is a comprehensive and well-designed study that addresses an important challenge in spatial multi-omics data integration. The authors present MAGPIE, a novel computational framework that is both generalisable and scalable, as convincingly demonstrated on datasets from multiple technologies and tissues. The application to a pharmacologically relevant model of lung injury effectively highlights the method's utility in deriving biologically and clinically meaningful insights.

In the revised manuscript, the authors have adequately addressed the reviewers' comments and have satisfactorily improved the clarity of the methodology and the discussion of the results. The revisions have strengthened the paper overall.

We thank the reviewer for their positive assessment and thoughtful feedback. We are pleased that the revisions improved the clarity and presentation of the work, and we appreciate the reviewer's supportive comments on the utility and robustness of MAGPIE.